# The cofactor-dependent folding mechanism of *Drosophila* cryptochrome revealed by single-molecule pulling experiments

Sahar Foroutannejad [1], Lydia L. Good[1], Changfan Lin [2], Zachariah I. Carter[3], Mahlet G. Tadesse [4], Aaron L. Lucius[3], Brian R. Crane [2] & Rodrigo A. Maillard [1]✉

The link between cofactor binding and protein activity is well-established. However, how cofactor interactions modulate folding of large proteins remains unknown. We use optical tweezers, clustering and global fitting to dissect the folding mechanism of *Drosophila* cryptochrome (dCRY), a 542-residue protein that binds FAD, one of the most chemically and structurally complex cofactors in nature. We show that the first dCRY parts to fold are independent of FAD, but later steps are FAD-driven as the remaining poly-peptide folds around the cofactor. FAD binds to largely unfolded intermediates, yet with association kinetics above the diffusion-limit. Interestingly, not all FAD moieties are required for folding: whereas the isoalloxazine ring linked to ribitol and one phosphate is sufficient to drive complete folding, the adenosine ring with phosphates only leads to partial folding. Lastly, we pro-pose a dCRY folding model where regions that undergo conformational transitions during signal transduction are the last to fold.

Since the solution of the first protein structures[1–3] much progress has been made to elucidate experimentally or computationally the tertiary structure and quaternary organization of proteins[4–7]. However, deter-mining protein folding pathways, i.e., identifying the steps and rate constants that connect transient intermediates to end states, has proven more challenging, particularly for large or multidomain proteins[8]. This challenge is further exacerbated when considering that a large fraction of the proteome incorporates cofactors, which are not only important for protein function but can also alter fold and ther-modynamic properties[9–11].

Previous studies in bulk have investigated the role of cofactor binding on the folding of flavoproteins that contain the cofactor flavin mononucleotide (FMN). These studies showed that flavoproteins can form molten globules or fold into their native state independently of the cofactor[12–15] or they can fold into an intermediate that is stabilized by FMN binding before reaching the native state[14]. More recently, single-molecule manipulation methods have enabled the direct kinetic characterization of intermediates along the folding pathway of small, single-domain proteins that have metal cofactors[16–20] However, these methods have not been applied to proteins that bind to flavin cofac-tors, let alone multidomain proteins that bind one of the most com-mon large cofactors known, flavin adenosine diphosphate (FAD)[9]. Single-molecule techniques like optical tweezers overcome several challenges in studying the folding mechanism of large proteins with bulk solution methods[21,22]. For instance, single-molecule methods enable the direct observation of transient intermediates, overcome the rapid loss of synchronicity among molecules undergoing sequential kinetic steps, provide a direct measure of changes in protein second-ary and tertiary structures, and avoid protein aggregation[23–27].

*Drosophila* cryptochrome (dCRY) is an FAD-binding multidomain protein of 542 residues that undergoes functionally important con-formational changes in response to chemical changes in FAD[28–30]. Thus, dCRY function is fundamentally linked to the coupling between the protein fold and the FAD cofactor. The structure of full-length

[1]Department of Chemistry, Georgetown University, Washington, DC, USA. [2]Department of Chemistry & Chemical Biology, Cornell University, Ithaca, NY, USA. [3]Department of Chemistry, University of Alabama at Birmingham, Birmingham, AL, USA. [4]Department of Mathematics and Statistics, Georgetown University, Washington, DC, USA. ✉e-mail: rodrigo.maillard@georgetown.edu

dCRY displays two large domains, collectively known as the photolyase homology region (PHR; Fig. 1a)[31–33]. The N-terminal domain comprises an α/β Rossman fold (residues 1-140)[34], whereas the much larger C-terminal domain (residues 141-542) contains a central 4-helix bundle motif conserved by other DNA-metabolizing enzymes that bind cofactors (residues 361-424)[35], as well as three extended loops that surround the FAD-binding pocket: (1) the phosphate-binding loop (residues 249-263) that coordinates a phosphate group close to FAD; (2) the protrusion motif (residue 288-306); and (3) the C-terminal lid (residues 420-446). These three loops, called C-terminal coupled motif (CCM), pack against a 10-residue helical tail called C-terminal tail or CTT (residues 530-539) (Fig. 1a)[31,32]. Signaling mediated by dCRY involves displacement of CTT from the CCM upon changes in FAD redox state[28–30].

In this work, we use optical tweezers to show that dCRY folds into its native state in a stepwise fashion via five intermediate states. We determined all the rate constants connecting these intermediates using clustering, bootstrapping and global fitting analysis. We find that FAD binds to the first two intermediates that are largely unfolded, yet the cofactor binds very fast (above the diffusion-limit) and with sub-nanomolar affinity. It is only after FAD binds and stabilizes these intermediates that dCRY can proceed to the native state. Furthermore, by using a variety of cofactors that contain different FAD moieties, we show that the isoalloxazine ring linked to ribitol and one phosphate

group (i.e., FMN) is sufficient for dCRY to natively fold. In contrast, the adenosine ring of FAD with one or two phosphate groups (i.e., AMP or ADP) causes formation of partially folded structures. By combining the results from optical tweezers experiments with published high-resolution structural data[31,32], we propose a model in which the dCRY PHR folds first and independently of FAD, followed by several steps of binding and co-folding around the cofactor by the larger C-terminal domain. Thus, the complex topology and domain organization of dCRY seems to require various folding strategies previously seen as mutually exclusive mechanisms for single-domain proteins that bind organic or metal cofactors. Altogether, our single-molecule approach allowed us to dissect and quantitate these distinct folding mechanisms for a single protein, underscoring the power and broad applicability of optical tweezers to dissect complex coupling mechanisms between folding of large proteins and cofactor binding.

## Results
### Mechanical unfolding trajectories of dCRY
To immobilize a single dCRY protein between the two beads in the optical tweezers (Fig. 1b), we added the Avi and ybbR tags at the N- and C-termini, respectively[36,37]. The Avi tag was covalently modified with biotin, and the ybbR tag was modified with a 350-base pair DNA handle with a digoxigenin in its 5′-end (Supplementary Fig. 1, Methods). The biotin interacts with a streptavidin-coated bead (SA bead) held in a

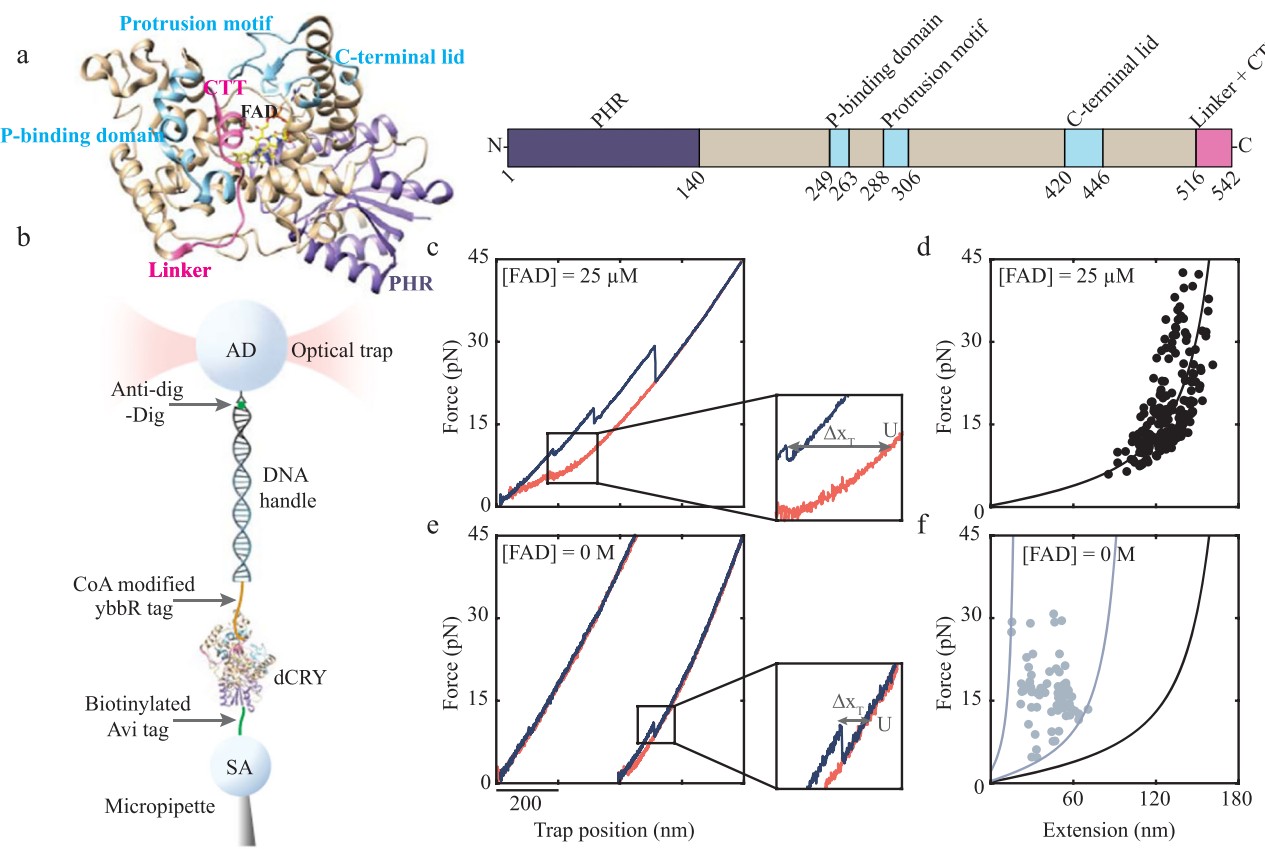

**Fig. 1 | Experimental design to study dCRY folding with optical tweezers.**
**a** Structure of full-length dCRY (left) and domain organization (right) (PDB ID: 4GU5). **b** Schematic representation of the optical tweezers set up to mechanically manipulate a single dCRY molecule between a streptavidin-coated bead (SA) and a bead coated with antibodies against antidigoxigenin (AD). **c** Force-extension trajectory of the mechanical unfolding (blue) and refolding (red) of dCRY in the presence of [FAD] = 25 μM. Indistinguishable results were obtained with and without Mg²⁺ at 0.5 mM. Zoomed-in is the total change in extension (ΔxT) measurement from the folded to the unfolded state. **d** Worm-like chain (WLC) analysis of ΔxT vs.

force of dCRY with [FAD] = 25 μM (blue dots). The solid line corresponds to the WLC model for full-length dCRY with a contour length of 204 nm and folded distance of 5.5 nm (Methods). **e** Force-extension trajectories of dCRY in the absence of FAD with no unfolding rips (left) and partially folded structures (right). Color code is as in (**c**). Zoomed-in is the unfolding event. **f** WLC analysis of ΔxT vs. force for unfolding of the protein in the absence of FAD (gray dots) with three WLC curves corresponding to contour lengths of 27 nm and 120 nm (gray lines) and 204 nm (black line).

fixed position on a micropipette tip. The digoxigenin binds a bead coated with antidigoxigenin antibodies (AD bead), which is held in a movable optical trap (Fig. 1b). Importantly, the modified protein displayed the same spectroscopic properties as wild type dCRY, indicating that the addition of the tags did not interfere with the normal fold of dCRY and its ability to bind FAD (Supplementary Fig. 2, Methods).

Molecular trajectories of the mechanical unfolding of single dCRY protein molecules were obtained by moving the bead in the optical trap away from or towards the bead on the micropipette tip. The resulting force-extension curve displayed a gradual increase in force and position, due to stretching of the DNA handle, that is interrupted by one or more rips that correspond to protein unfolding events (Fig. 1c, blue line). In this study, dCRY was mechanically unfolded using a constant pulling velocity of 75 nm/s and a force up to 45 pN. Then, the force was reduced to 1 pN to allow dCRY to refold and bind FAD before the next pulling cycle (Fig. 1c, red line). Depending on the type of experiment, we varied the refolding time between 0 and 40 s, the FAD concentration, or both.

## dCRY requires FAD to attain its native state

We first examined whether dCRY was natively folded in the single-molecule mechanical assay by determining the total change in contour length, $\Delta Lc_T$, upon unfolding, which reports on the number of folded residues at the start of the experiment. $\Delta Lc_T$ was obtained from the total unfolding rip size ($\Delta x_T$) irrespective of the number of intermediate rips that occurred during the unfolding reaction (Fig. 1c, inset). We used the worm-like chain (WLC) model[38] to analyze the observed $\Delta x_T$ as a function of force (Fig. 1d). In conditions where [FAD] = 25 µM and the refolding time was 20 s, we obtained a $\Delta Lc_T = 206 \pm 1$ nm (mean ± standard error, N = 269), which is consistent with the value of 198 nm that is expected from the fully folded dCRY structure[31,32] (see Analysis of Single-Molecule Trajectories in Methods). This result indicates that dCRY can reversibly fold, bind FAD, and attain its native folded state in the optical tweezers assay.

To study the role of FAD binding on dCRY folding, we tested if the protein could fold into its native state in the absence of the cofactor (Fig. 1e). Using the same refolding time of 20 s but with no FAD added into the optical tweezers microfluidic chamber, we found that 25% of force-extension curves did not display any unfolding rips, indicating that the protein remained unfolded, or sampled the unfolded state. The other 75% of the force-extension curves displayed small unfolding rip sizes (Fig. 1e, N = 132) with corresponding $\Delta Lc_T$ values between 27 nm and 120 nm (Fig. 1f). These values are significantly smaller than the observed $\Delta Lc_T$ for the natively folded protein bound to FAD (198 nm) indicating that dCRY can only form partially folded intermediates in the absence of the cofactor. Thus, to attain its native folded state, dCRY strictly requires a bound FAD molecule.

## Single-molecule FAD titration reveals multiple intermediates during dCRY folding

Given the essential role of FAD in driving the native fold of dCRY, we investigated how dCRY folding depends on FAD concentration. For this experiment, we monitored $\Delta Lc_T$ at FAD concentrations between 1 pM and 25 µM, allowing the protein to refold at 1 pN for 20 s between pulling cycles. The results were plotted as degree of folding, which was calculated by dividing the observed $\Delta Lc_T$ at each FAD concentration by 198 nm, the theoretical $\Delta Lc_T$ of full-length dCRY (Fig. 2a, see Analysis of Single-Molecule Trajectories in Methods). Degree of folding of 0 reflect the unfolded state, whereas values ~1 reflect the fully folded state. Degree of folding between 0.1 and 0.9 were considered intermediate states with partially folded structures.

The plots of degree of folding reveal a complex refolding pattern, with events that correspond to unfolded, intermediate or native states, and whose populations vary with FAD concentration (Fig. 2a). The fraction of events with degree of folding near 0 almost completely

disappeared as the FAD concentration was increased. In contrast, events with degree of folding around 1 increased gradually with FAD concentration. Events with degree of folding between 0.1 and 0.9 were observed throughout all FAD concentrations but with diverse degrees of folding (Fig. 2b). Assuming that the fully folded protein (degree of folding > 0.9) is bound to FAD, as it is shown in the X-ray structure (Fig. 1a), we plotted the fraction of fully folded states as a function of the log[FAD] to obtain an apparent FAD dissociation constant: $K_{d,app} = 0.11 \pm 0.01$ nM (mean ± standard error) (Fig. 2b).

## A complex dCRY folding pathway is coupled to FAD binding

The wide-ranging values of degree of folding observed in the FAD titration suggest that folding of dCRY coupled to FAD binding proceeds via multiple intermediate states. We therefore performed experiments to determine the kinetic steps and rate constants that connect these intermediates from the unfolded to the native state, including where in the folding pathway FAD binding occurs. For these experiments, we mechanically unfolded dCRY and allowed the protein to refold at 1 pN for increasing time intervals between 0 and 40 s, using concentrations of FAD of 0, 0.3 nM ($~K_{d,app}$) and 10 nM ($\gg K_{d,app}$). At each time interval and FAD concentration, we calculated the degree of folding from 0 and 1 (Supplementary Fig. 3, Methods).

To determine the number of states along the FAD-dependent folding pathway of dCRY, we applied model-based clustering to all the kinetic refolding data without the "zero values" (i.e., unfolded state) and used the Bayesian Information Criterion (BIC)[39,40] to determine the number of clusters (Supplementary Fig. 4a, Methods). This led to the identification of four clusters. Considering the "zero values" that represent the unfolded state as a separate cluster, the analysis results in five clusters (labeled C0 through C4) with degrees of folding centered at 0, 0.26 ± 0.01, 0.52 ± 0.01, 0.80 ± 0.01 and 1.0 ± 0.01 (mean ± standard error) (Supplementary Table 1). The five clusters identified by the statistical analysis are distinctively observed in the density plots[41] of the combined refolding data at each FAD concentration (Fig. 3a–c). We used bootstrapping to obtain standard errors for the component parameter estimates, including each cluster's mean, variance and weight (i.e., proportion of samples allocated to each cluster or fractions of dCRY states) (Supplementary Fig. 4b, Methods). The estimated fractions ± standard errors were plotted as a function of refolding time and FAD concentration (Fig. 3d–f) and globally fitted to various kinetic folding models (Supplementary Table 2, Methods).

Because clusters C0 and C4 with degree of folding 0 and 1 represent the unfolded state and the natively folded, FAD-bound structure, respectively, the remaining clusters C1, C2 and C3 must represent intermediate states. The cluster analysis does not identify whether FAD-bound and unbound dCRY states are incorporated in any of the five clusters. It is possible that all clusters include FAD-bound and unbound states, or just a few states bind the cofactor. However, the experimental data in Fig. 3a provides evidence that eliminates possible states along the dCRY folding pathway. For instance, the data at [FAD] = 0 shows that dCRY can only be allocated to clusters C0, C1 and C2, indicating that clusters C3 and C4 cannot form without FAD. Moreover, given the extensive intermolecular interactions established between folded dCRY and FAD[31,32] it is unlikely that the unfolded state binds FAD with high affinity, an observation that is in agreement with the result that the unfolded state is the only cluster that completely disappears with increasing FAD concentration (Fig. 2b). Thus, the resulting dCRY folding model includes clusters C1 and C2 as states that may bind FAD (Fig. 3g).

In this model, dCRY in the unfolded state (C0 = U) forms a first apo intermediate that can bind FAD (C1 = $I_1 + I_{1\cdot FAD}$). A second apo intermediate is formed from $I_1$, which also can form a complex with FAD (C2 = $I_2 + I_{2\cdot FAD}$). A third FAD-bound intermediate (C3 = $I_{3\cdot FAD}$) is formed from $I_{2\cdot FAD}$ before reaching the final FAD-bound native state

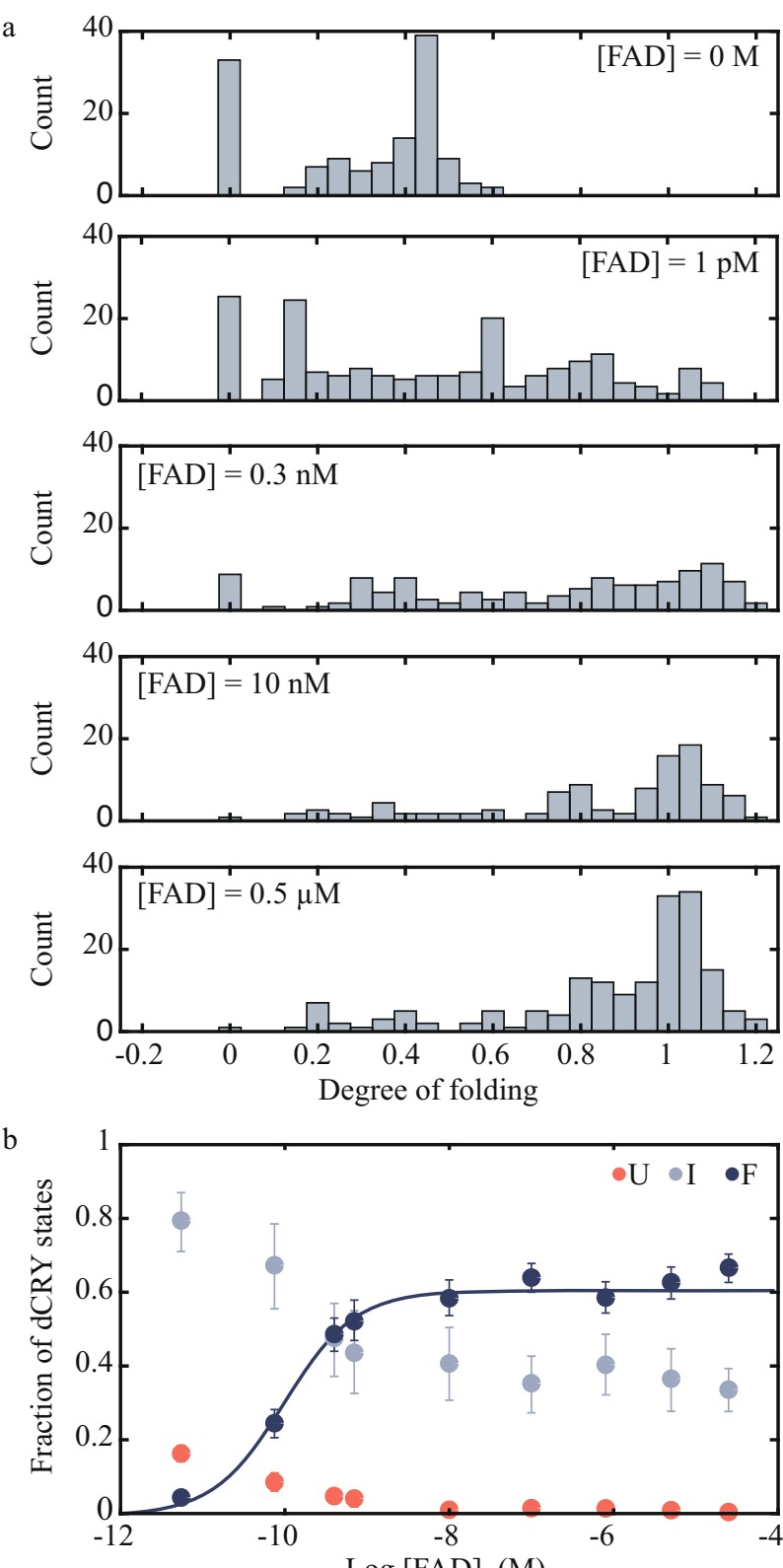

**Fig. 2 | Single-molecule FAD titration. a** Histograms of population of dCRY states vs. degree of folding in the presence of varying FAD concentrations. Unfolded, intermediate and folded states are shaded in light red, light blue and gray.
**b** Fraction of dCRY states (U unfolded, I intermediates, and F folded) plotted as a function of FAD concentration. The folded fraction, F, which is assumed to be bound to FAD, was fitted to a single site binding isotherm to obtain an apparent

dissociation constant, $K_{d,app} = 0.11 \pm 0.01$ nM (Section Analysis of Single-Molecule Trajectories in Methods). ($N = 6$ biologically independent samples per FAD concentration (i.e., 6 different single molecules evaluated), for a total of 3039 data points. Data are presented as mean values $\pm$ SD (standard deviation). Source data are available as a Source Data file.

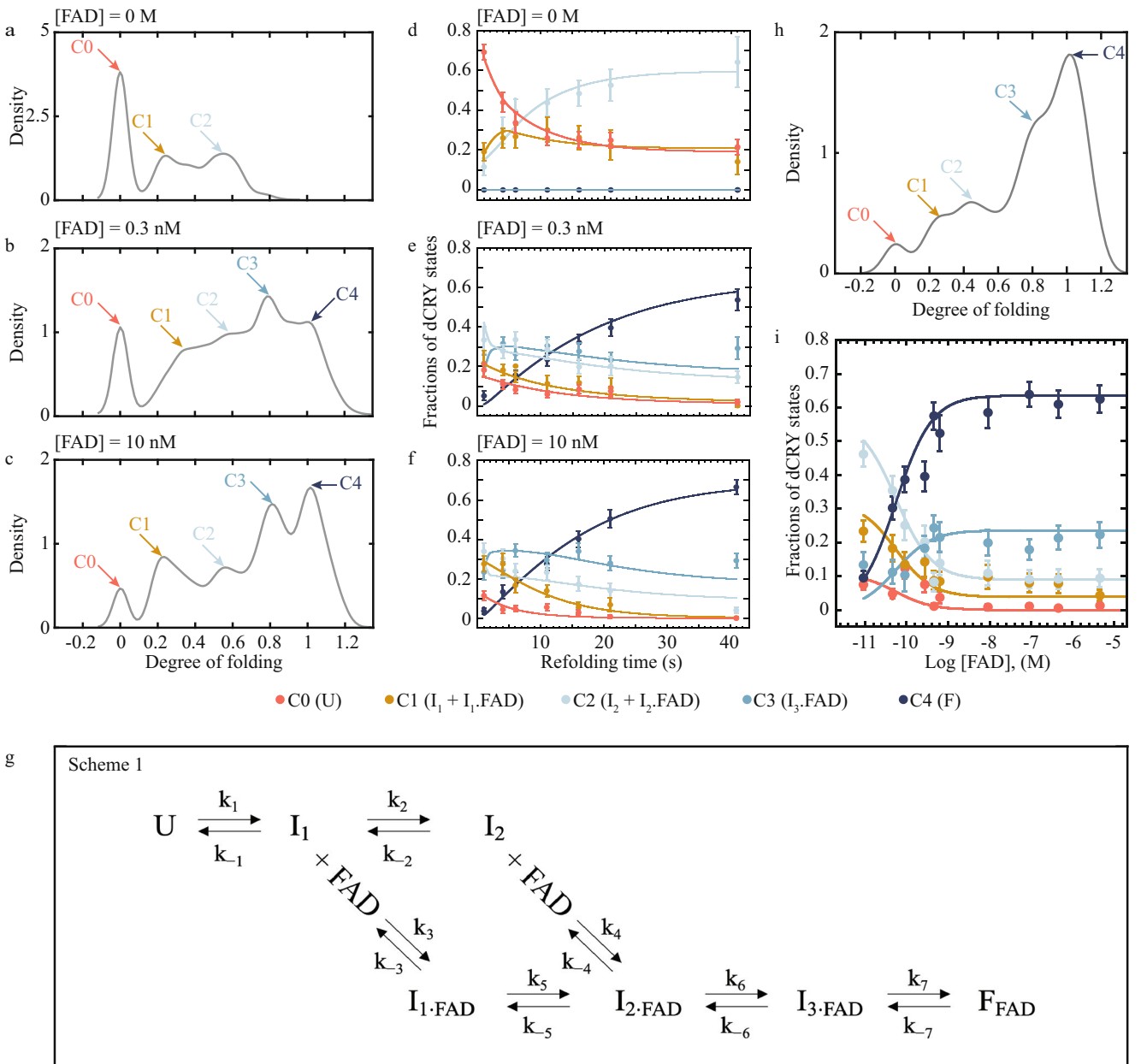

**Fig. 3 | Kinetic analysis of FAD binding coupled to dCRY folding. a–c** Density plots of degree of folding of combined data at [FAD] = 0, 0.3 nM and 10 nM, respectively. The combined density plots reveal five distinct clusters (labeled C0 thru C4) with degree of folding centered at 0, 0.26 ± 0.01, 0.52 ± 0.01, 0.80 ± 0.01 and 1.0 ± 0.01, respectively (Supplementary Table 1). **d–f** Plots of bootstrapped refolding data as a function of time at each FAD concentration. The data were globally fitted following the mechanism shown in the scheme in (**g**). **g** Model of dCRY folding. **h** Density plot of combined FAD titration data displays the same five cluster centers as in the kinetic refolding data shown in (**a–c**). **i** FAD titration plot of clustered and bootstrapped data from FAD titration experiments. The solid lines represent a global fitting of the scheme in (**g**). Fitted parameters are listed in Supplementary Table 2. Density plots in (**a–c, h**) provide a probability density estimate for each data using kernel density estimation with a Gaussian kernel and the optimum bandwidth[41]. The y-axis of (**d–f, i**) labeled Fractions of dCRY states correspond to the populations of each of the clusters normalized to 1. Data in (**d–f, i**) are presented as mean values ± SD. For each FAD concentration at each refolding time (**d–f**), 6 different single molecules (i.e., biologically independent samples) for a total of 2926 data points. For (**i**), $N = 6$ biologically independent samples per FAD concentration (6 different single molecules investigated), for a total of 3039 data points. Source data of (**d–f, i**) are available as a Source Data file.

(C4 = $F_{FAD}$). We also performed clustering and bootstrapping to the FAD titration data (Supplementary Table 1). The clustering analysis and the density plot of the combined FAD titration data (Fig. 3h) revealed the same five cluster centers obtained in the kinetic refolding experiments (Fig. 3a–c). The kinetic parameters in Fig. 3g were globally optimized to describe the time courses for the five clusters at the three concentrations of FAD as well as the FAD titration data (Fig. 3i). Binding and conformational equilibrium rate constants were determined between each state, first examining two simpler models in which FAD

only binds to $I_1$ or $I_2$, but not both. The resulting chi-squared tests indicate that the model where FAD binds to both $I_1$ and $I_2$ is statistically in better agreement with the data compared to the simpler model where FAD binds only to $I_1$ ($p$ value = 0) or $I_2$ ($p$ value < 0.1) when considering the FAD titration data combined with the kinetic refolding data (Supplementary Table 3, Methods). We fit the model in Fig. 3g to the kinetic refolding data or the FAD titration data by themselves and obtained rate constants that are in agreement with the combined dataset (Supplementary Table 2).

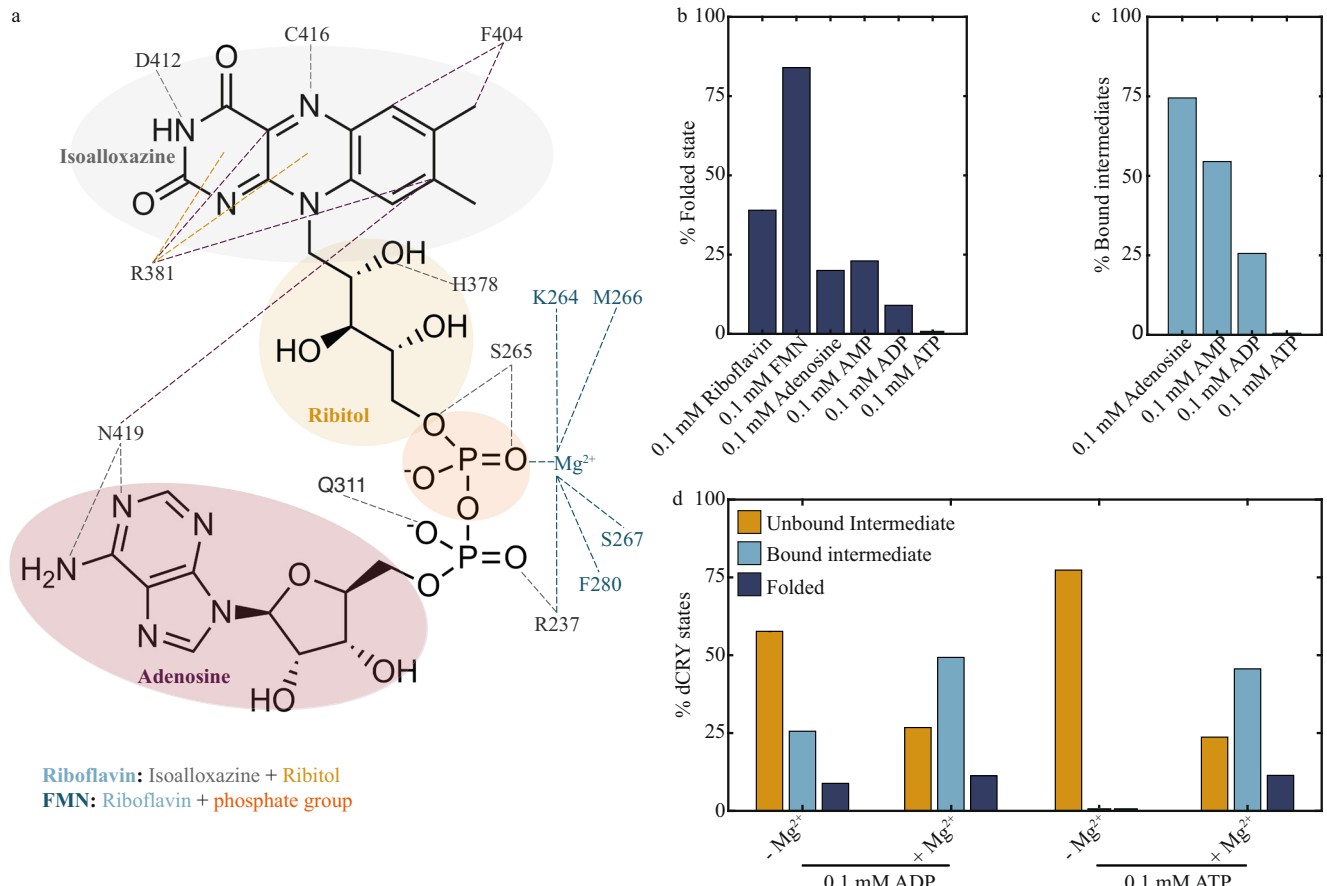

**Fig. 4 | Contribution of FAD moieties to dCRY folding. a** Contact map between FAD moieties and amino acid residues in dCRY. Interactions include hydrogen bonding (gray), hydrophobic effect (purple), metal coordination (dark teal) and cation-pi interactions (ochre). **b** Bar plot representing the percentage of dCRY in the folded state in the presence of different cofactors. **c** Bar plot of percentage of dCRY intermediate states bound to adenosine, AMP, ADP and ATP. **d** Bar plots of percentage of dCRY in an unbound intermediate, bound intermediate or folded state in the presence of ADP or ATP, with and without $Mg^{2+}$. Source data for panels **b**–**d** are available as a Source Data file.

The results from the global fitting allowed us to determine the fourteen rate constants that characterize the refolding pathway of dCRY (Fig. 3g). We find that $I_1$ is reversibly connected to $U$ with rate constants $k_1 = 0.15 \pm 0.03\ s^{-1}$ and $k_{-1} = 0.09 \pm 0.03\ s^{-1}$. The formation of $I_2$ is also reversible with $k_2 = 0.16 \pm 0.03\ s^{-1}$ and $k_{-2} = 0.06 \pm 0.02\ s^{-1}$. The reversibility of these two steps along the dCRY folding pathway is consistent with the refolding plot at [FAD] = 0 that shows co-existence between $U$, $I_1$, and $I_2$ at the longest refolding time intervals (Fig. 3d). In contrast to the relatively slow forward rate constants of formation of $I_1$ and $I_2$, the FAD binding rate constants for these two intermediates are substantially faster and comparable in magnitude with $k_3 = (2.8 \pm 0.4) \cdot 10^9\ M^{-1} \cdot s^{-1}$ and $k_4 = (10 \pm 0.9) \cdot 10^9\ M^{-1} \cdot s^{-1}$, respectively. The FAD unbinding rate constants for $I_1$ and $I_2$ are $k_{-3} = 0.7 \pm 0.3\ s^{-1}$ and $k_{-4} = 2.9 \pm 0.5\ s^{-1}$, respectively, resulting in equilibrium dissociation constants ($k_{-3}/k_3$ and $k_{-4}/k_4$) of 0.25 nM and 0.29 nM, which are in quantitative agreement with the apparent dissociation constant obtained in the single-molecule titration experiments (Fig. 2b). The conformational rate constant between $I_{1 \cdot FAD}$ and $I_{2 \cdot FAD}$ are $k_5 = 0.20 \pm 0.04\ s^{-1}$ and $k_{-5} = 0.09 \pm 0.06\ s^{-1}$, resulting in an equilibrium constant of 2.2 that favors the formation of folding intermediates with a larger degree of folding. After the formation of $I_{2 \cdot FAD}$ (either from FAD binding to $I_2$ or via a conformational change from $I_{1 \cdot FAD}$) a third FAD-bound intermediate, $I_{3 \cdot FAD}$, is formed with $k_6 = 0.9 \pm 0.1\ s^{-1}$ and $k_{-6} = 0.38 \pm 0.08\ s^{-1}$. The fitted parameters for the formation and disappearance of the fully folded state, $F_{FAD}$, are $k_7 = 0.12 \pm 0.04\ s^{-1}$ and $k_{-7} = 0.044 \pm 0.004\ s^{-1}$. Using the rate constants from Fig. 3g, we determined the average folding time from U to $F_{FAD}$ as a function of

FAD concentration (Supplementary Fig. 5, Methods). At conditions where [FAD] >> $K_d$, the average folding time plateaus at 30 s. Using this value, we calculated that the probability of dCRY reaching $F_{FAD}$ in 40 s is 0.73 (Methods), which is consistent with our experimental observations showing that the fraction of $F_{FAD}$ when the protein refolds for 40 s at [FAD] = 10 nM is $0.66 \pm 0.4$ (Fig. 3i).

Altogether, the statistical and kinetic analyses of the data indicate that dCRY folding is slow and follows a complex pathway, wherein FAD binds fast to early intermediates with partially folded structures. These early intermediates likely represent a minimal structural scaffold for FAD docking to promote the natively folded FAD-bound state. Considering the experimentally determined FAD concentrations in eukaryotes, which range between 8-240 nM[42], binding of FAD to the intermediates $I_1$ and $I_2$ contributes the most to the stability of native state, $F_{FAD}$.

## Contribution of FAD moieties to dCRY folding

To further dissect the mechanism by which FAD promotes the native fold of dCRY, we investigated how the different FAD moieties contribute to dCRY folding. FAD harbors two ring structures that are connected by two phosphate groups in tandem (Fig. 4a). The first ring structure is riboflavin, which is composed of the isoalloxazine ring covalently linked to ribitol, a pentose. The other ring is adenosine, composed of ribose and adenine. These moieties have different properties that establish hydrophobic, ionic, and hydrogen bond interactions with dCRY (Fig. 4a). In addition, the two phosphate groups in FAD interact with an $Mg^{2+}$ ion. We used optical tweezers to

determine the degree of folding of dCRY using various FAD moieties. As in previous experiments, the protein is allowed to refold for 20 s at 1 pN between pulling cycles.

We found that riboflavin (at 0.1 mM) is able to promote a fully folded dCRY structure with a degree of folding of $0.99 \pm 0.01$ nm in 39% of events. In the other 61% of events, we obtained a degree of folding of $0.78 \pm 0.01$ ($N = 256$) (Fig. 4b, Supplementary Table 4). Interestingly, events corresponding to unfolded or intermediate states with a degree of folding below 0.3 were negligible, indicating the absence of protein in the apo state (Supplementary Fig. 6). The high percentage of riboflavin-bound intermediates may be due to the fact that riboflavin cannot establish all the intermolecular interactions that FAD allows. We therefore used FMN, composed of riboflavin and one phosphate group, to determine the contribution of the phosphate to dCRY folding. At 0.1 mM of FMN, most events displayed a degree of folding of 1 (84%, $N = 158$, Fig. 4b, Supplementary Fig. 6). This result indicates that the phosphate group next to riboflavin is critically important for dCRY folding. In agreement with this observation, the dCRY structure shows that this phosphate group establishes two hydrogen bonds with Ser265 (Fig. 4a), located in the phosphate-binding motif of the protein.

The results obtained with riboflavin and FMN indicate that the isoalloxazine ring is a major contributor to dCRY folding. It may seem, therefore, that the role of the adenosine ring in promoting and reaching the native conformation is minor. We examined the contribution of adenosine (at 0.1 mM) to dCRY folding and observed a wide distribution of intermediates with degree of folding from 0.3 to 0.9 in 75 % of events ($N = 286$), and the fully folded conformation in 20% of events (Fig. 4c, Supplementary Fig. 6). Thus, while the adenosine ring is able to promote complete folding of dCRY, it does so much less efficiently compared to the isoalloxazine ring in riboflavin or FMN. Given that the adenosine ring establishes contacts with residues in dCRY that are different from the isoalloxazine ring (Fig. 4a), it is possible that the folding pathway is going to be different, or the kinetic rates are going to be slower for adenosine-containing cofactors due to their lower folding efficiency.

## Burial of FAD phosphates is energetically costly but required for efficient dCRY folding

Because the phosphate group in FMN was found to be critically important in promoting a folded conformation, we investigated the presence of one or two phosphate groups in adenosine by using AMP and ADP, respectively. Whereas the distribution of species was similar between adenosine and AMP (Supplementary Fig. 6), the presence of two phosphates decreased the percentage of fully folded protein from 24% with AMP to 9% with ADP ($N = 327$ for AMP and $N = 215$ for ADP, Fig. 4c). Another important effect owed to the second phosphate in ADP is in the observed intermediate states. The events with degree of folding between 0.5 and 0.9, that reflect bound intermediates based on the kinetic refolding experiments, were much smaller in ADP than in AMP or adenosine (Fig. 4c). These results indicate that the second phosphate group has opposing effects depending on whether it is covalently linked to the adenosine ring in ADP (negative folding effects) or to the isoalloxazine ring in FMN (positive folding effects).

Since charges are characteristically incompatible with hydrophobic environments, and only highly stable proteins are found to tolerate engineered charged residues in their hydrophobic core[43], it is possible that the negative effect of ADP on dCRY folding is due to the energetic penalty of partially burying phosphate charges in the protein core. We tested this hypothesis by increasing the number of phosphate groups in adenosine by using ATP. When ATP is 0.1 mM, we observed that dCRY samples the unfolded state (23%, $N = 128$) or forms partially folded intermediates (77%) with degree of folding of up to ~0.5 (Supplementary Fig. 6). No events corresponding to the natively folded state or intermediates with degree of folding > 0.5 were observed

(Fig. 4b, c). Similar percentages and degrees of folding were obtained in experiments with no FAD, suggesting that ATP at 0.1 mM may not bind to dCRY or can bind and only form early intermediates. Thus, the results with ATP support our interpretation that increasing the number of negative charges has unfavorable effects on the ability of the adenosine ring to promote the native state of dCRY.

It is possible, however, that the additional phosphate group in ATP may not only increase the number of negative charges per adenosine molecule but also generate steric hindrance effects. If steric hindrance were a major force in preventing ATP from interacting and promoting folding in dCRY, then our previous interpretation would be invalid. Therefore, we performed similar experiments with ATP at 0.1 mM but in the presence of saturating $MgCl_2 = 0.5$ mM to reduce the net negative charge of the molecule[44–46]. If $ATP-Mg^{2+}$ and ATP alone have similar effects on dCRY folding, then steric hindrance may be the dominant force in inhibiting ATP to interact and induce dCRY to fold. We found, however, that the percentage of events corresponding to intermediates with degree of folding below 0.5 reduces to 24% with $Mg^{2+}$ from 77% with no $Mg^{2+}$ (Fig. 4d). And, accordingly, the fraction of events with degree of folding between 0.5 and 0.8, corresponding to ATP-bound intermediates, increases from 0 to 45%. Only 11 % of events corresponded to the fully folded state with degree of folding larger than 0.9 ($N = 114$). These observations indicate that the negative folding effects of the phosphate groups in adenosine can be mitigated by the presence of $Mg^{2+}$. In fact, the crystal structure of dCRY shows a single $Mg^{2+}$ ion interacting with both phosphate groups of ADP (Fig. 4a). It is therefore possible that the role of the $Mg^{2+}$ ion is to reduce the energetic penalty of burying a negative charge in FAD. In support of this observation, the addition of $MgCl_2$ (0.5 mM) to ADP (0.1 mM) also increases the fraction of ADP-bound intermediates, albeit to a lesser degree compared to ATP (Fig. 4d, Supplementary Table 4).

Interestingly, it is well-established that plant CRYs bind ATP and other nucleotides in a cavity close to FAD[47,48]. Binding of ATP to this secondary site increases the quantum yield of the signaling state, facilitates protonation of the flavin when it gets photoreduced, and has been proposed to stabilize conformational changes in the α/β domain important for signaling[49–54]. It is not known whether dCRY has the secondary binding site for ATP. Our studies show that dCRY-ATP interactions mostly induced the formation of partially folded intermediates. Further studies using mixtures of FAD and ATP may help to elucidate if dCRY has a secondary site for ATP that is important for function but less relevant for folding in the native state.

## Mapping folding and FAD-bound intermediates along the dCRY folding pathway

To gain greater insight into the folding pathway between U and $F_{FAD}$ we sought to identify possible structures for the intermediates that were identified in the kinetic refolding experiments (Fig. 3g). As in previous mechanical manipulation studies of single molecules[27,55–60] we estimated the intermediate structures by comparing the experimentally determined $\Delta Lc_T$ to values expected from folding different parts of the protein (Supplementary Fig. 7, Methods). This approach considers that loops and random coils are unstable against mechanical force, and that secondary structures are stable when they are intact or fully folded, i.e., intermediates ending in the middle of β-strands or α-helices are energetically disfavored[48,50]. For dCRY, the values of $\Delta Lc_T$ for the three intermediates were estimated from the cluster analysis and yielded mean degrees of folding of 0.26, 0.52, 0.80 for clusters C1, C2, and C3, respectively. These values correspond to 140 amino acids for cluster C1 composed of $I_1$ and $I_{1·FAD}$, 275 amino acids for cluster C2 composed of $I_2$ and $I_{2·FAD}$, and 448 amino acids for cluster C3 composed of $I_{3·FAD}$. (Supplementary Table 5, Supplementary Fig. 7).

The model we propose describes the unfolded polypeptide undergoing a stepwise refolding process to attain its native, FAD-

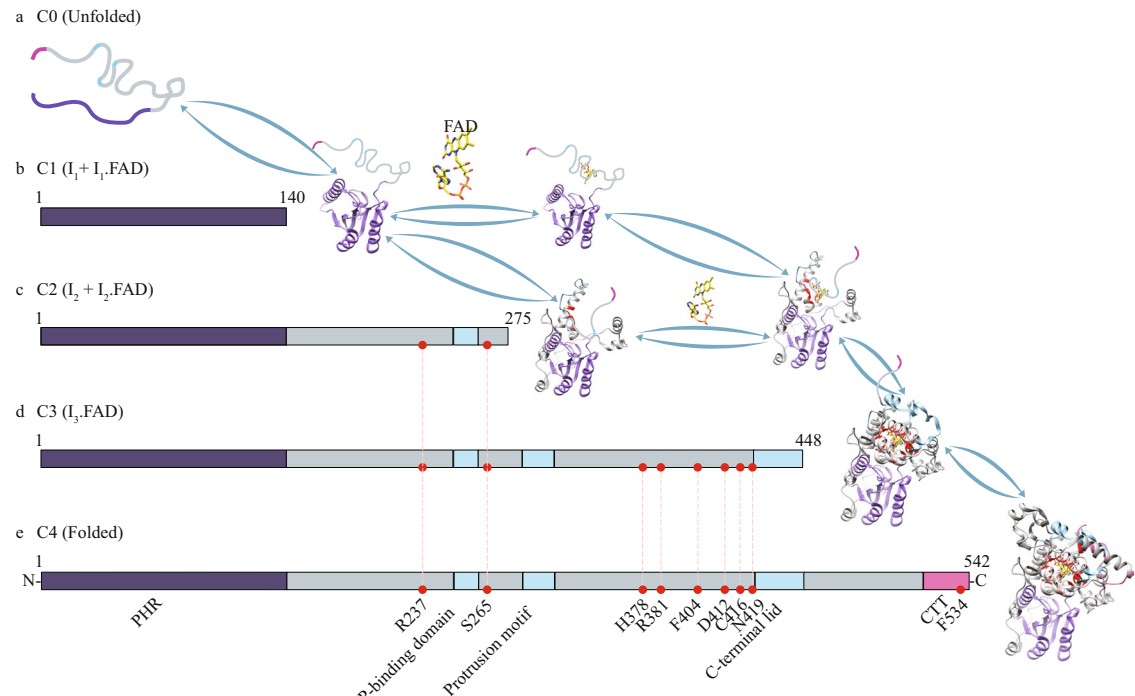

**Fig. 5 | Mapping intermediate structures and FAD-binding events along the dCRY folding pathway. a, b** The unfolded polypeptide folds into the first intermediate ($I_1$) that matches with the size of PHR (purple). **c** $I_1$ either binds FAD to form $I_1$·FAD or folds into $I_2$, which encompasses the Phosphate-binding motif. **d** $I_2$ can bind FAD to form $I_2$·FAD, which then folds into $I_3$·FAD that includes folding of the protrusion motif and the C-terminal lid. Together with the phosphate-binding motif, $I_3$·FAD has the C-terminal coupled motif (CCM) folded, which include the entire network of residues that interact with FAD. **e** In the last step, $I_3$·FAD folds into the dCRY native structure and the C-terminal tail (CTT) docks on top of FAD.

bound structure (Fig. 5). We find that cluster C1 ($I_1$ and $I_{1\cdot FAD}$) matches the size of 140 residues in the N-terminal α/β domain of the protein (Fig. 5b). Importantly, this is the only structural domain in dCRY that does not make any direct interactions with FAD, and our refolding studies show that this intermediate can fold in the absence of the cofactor (Fig. 3g). We propose that cluster C2 ($I_2$ and $I_{2\cdot FAD}$) incorporates the N-terminal domain up to residue Leu275 (Fig. 5c). This reasoning is based on the large number of unfolding trajectories in cluster C2 that include a short-lived transition of degree of folding of 0.26 that match cluster C1 or the PHR. Cluster C2 includes Arg237 and the phosphate-binding domain (residues 249-263), which make direct interactions with FAD (Fig. 4a). Among these residues, Arg237, Lys264, Ser265, and Met266 also interact with the magnesium ion observed in the dCRY structure[31,32].

Cluster C3 or $I_{3\cdot FAD}$ maps to residues starting from the N-terminus to Asp448, which harbor all the motifs that form the FAD-binding pocket in dCRY, including the conserved 4-helix motif common to photolyase and DNA primase[35], and the surrounding loops called the C-terminal coupled motif or CCM (Fig. 5d). The 4-helix motif binds the isoalloxazine ring and the CCM completes the interaction network with FAD through Phe280, Gln311, His378, Arg381, Phe404, Asp412, Val415, Cys416, and Asn419 (Fig. 4a). As stated for cluster C2, unfolding trajectories of cluster C3 also include short-lived transitions that matched the size of the PHR. After the formation of $I_{3\cdot FAD}$, the remaining residues required to attain the native state correspond to the C-terminal linker and the C-terminal tail (CTT). This step is expected to occur at the end of the folding pathway since the C-terminal linker and the CTT dock on top of FAD and in between the scaffold formed by the CCM (Fig. 5e). Moreover, the last step of folding of the CTT is consistent with functional studies that showed that signaling in dCRY involves the displacement of the CTT upon changes in flavin redox state[30]. Hence, its association with the protein core via the CCM is probably the last folding step of dCRY, consistent with our

analysis of optical tweezers experiments. Altogether, our single-molecule studies indicate that $I_{3\cdot FAD}$ is likely the first functional intermediate of dCRY, i.e., intermediates prior to $I_{3\cdot FAD}$ do not have the scaffolding to allow changes in conformation dependent on the FAD redox state.

## FAD binds to dCRY intermediates following two different mechanisms

Interestingly, both $I_1$ and $I_2$ bind to FAD with similarly high affinities (Supplementary Table 2), but their cofactor binding mechanisms are likely distinct. For instance, $I_1$ does not include folded residues that directly interact with FAD, while $I_2$ has preformed a first structural scaffold for direct contact with FAD. Cofactor interactions with $I_1$ seem to follow an induced-fit binding mechanism, where $I_{1\cdot FAD}$ may promote the formation of a better-defined FAD-binding site, as mapped for $I_{2\cdot FAD}$. Noteworthy, $I_1$ is almost 75% unfolded, including all residues involved in FAD binding, and cofactor binding still occurs with very fast kinetics ($2.8 \cdot 10^9\,M^{-1} \cdot s^{-1}$). Several binding studies involving intrinsically disordered proteins (IDPs) have reported association rates around the estimated diffusion limit for folded proteins ($10^9$–$10^{10}\,M^{-1} \cdot s^{-1}$)[51–55]. Such fast association kinetics for IDPs has been explained by "fly-casting" effects, where the unfolded polypeptide forms initial interactions with its binding partner at a greater distance than a folded protein[61–63]. It is plausible that the large fraction of unfolded polypeptide seen in $I_1$ binds FAD with fast kinetics following a fly-cast effect, as seen in IDPs. In contrast, cofactor interactions with $I_2$ likely follow a conformational selection mechanism[64], where FAD selects a state with a preformed, albeit not fully structured, binding pocket. Given the degree of burial of FAD in the fully folded dCRY structure, and the fact that dCRY dynamics are mostly localized to the CTT upon changes in flavin redox state[28,29], binding of FAD to intermediate states seems a more likely scenario, as opposed to a model in which FAD binds after a fully folded structure forms, which

would require large conformational changes or significant unfolding to then incorporate the cofactor.

Altogether, our results indicate that FAD binding to dCRY involves multiple mechanisms, as reported for other proteins[65,66], including proteins that bind FMN, a related cofactor[67]. It is possible that topologically complex proteins with slow folding kinetics as seen for dCRY have evolved to bind a cofactor to different intermediates along its folding pathway to prevent the formation of misfolded states that could accumulate and lead to protein aggregation.

## Discussion

A large fraction of proteins harbor inorganic (i.e., metal clusters) or organic (i.e., FMN, FAD, hemes, among others) cofactors in their structure. While the functional role of protein cofactors has been well described and characterized for decades, how cofactor interactions contribute to protein folding, conformation and stability is not well understood. Previous bulk and single-molecule studies have investigated how a cofactor interacts with small, single-domain proteins[16–20]. These studies have shown that a protein can either interact with the cofactor in the unfolded state, forming and stabilizing an intermediate before reaching the native state[68–70], or fold and reach the native state before binding the cofactor[18]. However, mechanistic studies on cofactor interactions coupled to folding for large proteins with multiple domains are lacking. Recently, the application of single-molecule optical tweezers has opened opportunities to study the folding mechanisms of large proteins or protein complexes[23–26,71]. Here, we used optical tweezers to study folding of dCRY, a multidomain protein of 542 amino acid residues that binds to one of the most common and complex organic cofactors, FAD.

The molecular trajectories of the unfolding behavior of dCRY under force displayed large heterogeneity, suggesting multiple intermediates along the folding pathway of dCRY (Fig. 1). Given the complexity of the unfolding trajectories, seen with and without FAD, we established a statistical framework based on clustering and bootstrapping procedures (Supplementary Figure 4) that enabled a quantitative characterization of such heterogeneous data. The data were then globally fit[72] to different models of folding coupled to FAD binding, which resulted in the quantification of the 14 binding and conformational rate constants that reversibly connect the 7 states shown in the dCRY folding mechanism (Fig. 3g). We note that the best fit parameters obtained in this study were obtained at 1 pN. While this force is close to zero pN, future experiments in which the refolding force is varied will be required to identify which kinetic rate constants may have a strong force dependence.

Given the high degree of heterogeneity of the single-molecule data, it is possible that the analysis used in this study missed transient states that have very low probabilities and therefore the number of clusters may be underestimated. Thus, the folding mechanism of dCRY may be more complex than what we have outlined. Nonetheless, the methods described herein allow for unprecedented quantification of key states in single-molecule folding data that display such a high degree of complexity and heterogeneity.

The folding mechanism of dCRY revealed multiple intermediates that can bind FAD and lead to the native FAD-bound structure. We find that FAD is strictly required for dCRY to attain its native state, i.e., without FAD the protein never folds to its biologically functional structure. This is in contrast to the mammalian clock CRYs that have a homologous structure, yet they fold without FAD. In fact, in the absence of FAD, dCRY only forms two intermediates with 26% and 52% of folded polypeptide. WLC analysis indicates that the second intermediate with 52% of folded polypeptide incorporates a minimal folded scaffold for FAD binding. However, the folded structures that map to the first intermediate correspond to the N-terminal α/β domain which does not establish direct contacts with the cofactor. Nonetheless, FAD binds to the first and second intermediates similarly fast, displaying

association rate constants above the diffusion limit established for globular proteins[73]. We propose that FAD binds to these intermediates following two different mechanisms: an induced-fit binding mechanism for the first intermediate, which has been reported for IDPs that bind ligands or protein partners following fly-casting effects[61], and a conformational selection binding mechanism[64] to the second intermediate, in which FAD selects an intermediate state with a preformed, yet precursory, cofactor binding pocket.

Given the large size of dCRY, it was expected that the unfolded polypeptide in the apo state samples a large number of heterogenous folding pathways, which would reduce in number as the protein progressively establishes native-like contacts[73]. Instead, the data revealed well-defined apo and FAD-bound pathways, with intermediates that share similar degrees of folding with and without the cofactor (Fig. 3g). These intermediates may represent cooperative units within the structure of dCRY, which for other proteins have been defined as foldons[74]. Our data, therefore, suggest that dCRY has a defined pathway that progressively folds and binds FAD. It is possible that FAD has evolved to stabilize pre-existing foldons, thereby decreasing the likelihood of misfolding while at the same time promoting intermediate states with larger degrees of folding until the native state is reached.

By studying the effect of various FAD moieties on the degree of folding of dCRY (Fig. 4a) we determined that the isoalloxazine ring with ribitol and one phosphate (i.e., FMN) is sufficient to drive the native state of the protein (Fig. 4b). In contrast, the adenosine ring covalently linked to one or two phosphates (i.e., AMP and ADP, respectively) does not promote native folding to the same degree, but mostly forms intermediates. Unexpectedly, the adenosine ring without the phosphates is more efficient than AMP or ADP at promoting folding (Fig. 4b, c). This finding suggests that the partial burial of negative charges by dCRY is energetically penalized. We show that $Mg^{2+}$ ions that coordinate to the phosphate of ADP overcome such energetic penalty, resulting in a higher percentage of bound intermediates or the folded state. We further investigated this model by determining the percent of fully folded events using ATP, with and without $Mg^{2+}$. We observed an increase in fully folded events when the ion is present (Fig. 4d). Altogether, this study dissected how the different moieties in FAD contribute to the folding of dCRY in a non-additive fashion. Moreover, the fact that various FAD moieties interact with dCRY and promote folding to varying degrees indicates that initial contacts between unfolded dCRY and FAD can be established with different moieties, increasing the probability of productive folding.

Interestingly, the structurally homologous mammalian clock CRYs fold and carry out their functions void of the cofactor[75,76]. There is experimental evidence in vitro that flavin binding is linked to their function, but it is not conclusive[77–79]. However, as a principal photoreceptor, we show that dCRY strictly requires FAD to fold into its functional structure. In this context, our studies allow us to speculate about the evolution of cofactor binding. It is possible that the protein interacted with the cofactor in an unfolded state, and then folded around the cofactor, as seen between FAD and dCRY. As the cofactor locked and optimized this interaction, the protein could fold without the cofactor as functions diverged, which is the case for mammalian CRYs. Thus, the structural template, or CRY fold, became stable without the need of the cofactor. Comparative single-molecule studies of folding of mammalian CRYs with and without FAD may help in better understanding how cofactor binding and folding are evolutionarily linked.

## Methods

### *Drosophila* cryptochrome (dCRY) modification, expression and purification

dCRY was co-expressed as a fusion protein with a His$_{6x}$ tag. For single-molecule study it was also engineered to have an Avi tag at the N-terminus and an ybbR tag at the C-terminus of the protein. The

protein was purified following previous protocols[80]. Briefly, cells were grown in Terrific Broth (IBI) at 37 °C and induced with 0.4 mM IPTG and 5 µM FAD after the optical density (OD) at 600 nm reached 0.6–0.8. Cells continued to grow overnight at 17 °C before sonication in lysis buffer containing 50 mM Hepes (pH 8), 400 mM NaCl, 10% glycerol (vol/vol), 0.5 mM TCEP, and 0.5% Triton X-100 (vol/vol). The cell lysate was centrifuged at 48,000 × $g$ for 1 h to remove cell debris. The supernatant was incubated with Nickel-NTA Agarose Beads, tehn cleaned with wash buffer containing 50 mM Hepes (pH 8), 400 mM NaCl, 10% glycerol (vol/vol), and 20 mM imidazole. dCRY was eluted in buffer containing 50 mM Hepes (pH 8), 100 mM NaCl, and 10% glycerol (vol/vol) under an imidazole gradient that varied from 40 to 200 mM. dCRY was further purified using a Superdex 200 size exclusion column in 50 mM Hepes (pH 8), 150 mM NaCl, 10% glycerol (vol/vol), and 2 mM TCEP. All DNA was sequenced at the Cornell Biotechnology Center.

### Oligonucleotide modification with acetyl-CoA
A single-stranded oligonucleotide modified by an amine group in the 5′ end (ssOligo, 5′-NH$_2$-CTGCTGGGGCAAACCAGCGTGGAC-3′) was reacted with SM(PEG)$_8$ cross-linker in a 1:60 molar ratio at room temperature for 3 h. Then, 50 mg/mL aqueous solution of acetyl-CoA was added in a 1:4.8 molar ratio to SM(PEG)$_8$ into the reaction mixture. The reaction was proceeded at room temperature overnight and quenched by addition of 5 µL of 1 M Tris pH 6.8 and 5 µL of 1 M β-mercaptoethanol. The product was isolated by electroelution, and ethanol precipitation followed by annealing to its reverse complementary sequence modified with a phosphate group in the 5′ end (5′-Phos-CGACGTCCACGCTGGTTTGCCCCAGCAG-3′). The resulting product named dsoligo-CoA has an overhang of 4 nucleotides for ligation (underlined sequence) (Supplementary Fig. 1).

### Protein modification
dCRY modification for single-molecule studies was performed first by biotinylating of the Avi tag followed by SFP-mediated covalent attachment of dsoligo-CoA to ybbR tag[81] (Supplementary Fig. 1). The dsOligo-CoA was further ligated to a digoxigenin-modified 350-base pair long DNA handle following protocols published in our laboratory[82]. Tagged dCRY was incubated in biotinylation buffer (25 µM D-biotin, 5 mM ATP, 5 mM Mg(OAc), pH 7.4) with BirA at a final concentration of 4 µM at 4 °C for 2 h. A biotin-streptavidin pull-down assay was performed to ensure biotinylation of the protein (Supplementary Fig. 1). SFP and dsOligo-CoA were added to the reaction in a 3:3:1 molar ratio of dCRY to SFP to dsOligo-CoA in 50 mM HEPES and 10 mM MgCl$_2$, at 4 °C for 6 h. Prior to each single-molecule experiment, a ligation reaction between modified dCRY with dsOligo-CoA and Dig-modified DNA handle was freshly performed in a 10 µL reaction[82] (Supplementary Fig. 1).

### Optical tweezers experiments
Optical tweezers experiments were performed in a MiniTweezers instrument[83] in dCRY buffer (50 mM Tris pH 8.0, 150 mM NaCl, 10 mM DTT) supplemented with the desired final concentration of FAD or other cofactors (see experimental details below). For experiments in the absence of FAD (i.e., apo conditions), we did not add any FAD to the protein sample and verified the absence of the cofactor by lack of any detection of FAD absorption using a UV spectrophotometer (Supplementary Fig. 2). For each condition, at least 6 different molecules were sampled for a total of >150 molecular trajectories. Force ramp experiments were performed at a constant pulling rate of 75 nm/s sampled at 200 Hz to a maximum unfolding force of 45 pN. Formation of a single tether was confirmed by overstretching of the DNA handle up to ~65 pN[38].

### dCRY folding as a function of FAD concentration
Modified dCRY was first dialyzed in dCRY buffer (50 mM Tris pH 8.0, 150 mM NaCl, 10 mM DTT) overnight at 4 °C. Before each experiment,

dCRY buffer was supplemented with the desired concentration of FAD to equilibrate the optical tweezers main chamber ranging between 0 to 25 µM. After forming a tether, the protein was mechanically unfolded and refolded as described above and allowed to refold for 20 s at 1 pN of force (Fig. 2a).

### Kinetic refolding experiments as a function of FAD concentration
For time-dependent refolding experiments, modified dCRY were prepared and mechanically unfolded and refolded as described above. By the end of each pulling cycle, the protein was allowed to refold for 1, 3, 5, 10, 15, 20, and 40 s. Each of these refolding time points were done in the presence of 0 M (<< K$_{d,app}$), 0.3 nM (~ K$_{d,app}$) and 10 nM (>> K$_{d,app}$) (Supplementary Fig. 3).

### Single-molecule experiments in the presence of FAD moieties
For each experiment, FAD was first dialyzed out from the protein sample and supplemented with a specific cofactor (i.e., riboflavin, FMN, adenosine, AMP, ADP, ATP) (Supplementary Fig. 6). The ligation reaction and dCRY buffer for optical tweezers main chamber were supplemented with the desired cofactor as well. The experimental final concentrations for all the moieties were adjusted to 0.1 mM of the cofactors. We also used ADP and AMP at 1 mM, as well as in the presence of 0.5 mM MgCl$_2$. In between pulling cycles, the protein was allowed to refold at 1 pN for 20 s.

### Analysis of single-molecule trajectories
The data from all molecular trajectories of unfolding-refolding of dCRY were parsed in MATLAB (R2021b). We selected individual trajectories of each unfolding-refolding cycle and measured the total change in extension ($\Delta x_T$) of the protein from the first observed unfolding rip to the unfolded state at the same force (Fig. 1c or e)[84]. The values of $\Delta x_T$ vs. force were analyzed using the Worm-like chain (WLC) model to obtain the total change in contour length upon unfolding, $\Delta Lc_T$ (Eq. 1)[38]:

$$F = \frac{k_B T}{p} \left[ \frac{1}{4} \left( 1 - \frac{x}{Lc} \right)^{-2} - \frac{1}{4} + \frac{x}{Lc} \right] \tag{1}$$

where $p$ is the persistence length of the chain ($p = 0.65$ nm used for polypeptides), $x$ is the end-to-end extension of the folded structure, and $Lc$ is the contour length (number of amino acids of the polypeptide multiplied by 0.365 nm/amino acid). The observed $\Delta Lc_T$ is obtained from the difference between the estimated $Lc$ and the end-to-end extension of the folded structure ($\Delta Lc_T = Lc - X_{Folded}$). $\Delta Lc_T$ reflects the amount of folded polypeptide that participated at the start of the unfolding reaction in the optical tweezers experiment. The expected value for the fully folded state is 558 amino acids of native protein and engineered linkers × 0.365 nm per amino acid−the folded end-to-end distance (5.5 nm) = 198 nm.

To quantitatively report the degree of folding of each molecular trajectory, we divided the observed $\Delta Lc_T$ with the value expected for natively folded dCRY, based on the crystal structure: 558 amino acids of the native protein and engineered linkers × 0.365 nm per amino acid − 5.5 nm corresponding to the folded end-to-end distance = 198 nm. Degree of folding enabled us to initially categorize the FAD titration data into three states: unfolded state (degree of folding ≤ 0.1), intermediate state (0.1 < degree of folding ≤ 0.9), and folded state (0.9 < degree of folding). This division is based on the observed standard deviation of $\Delta Lc_T$ of the fully folded protein, which is 15 nm or 7%. We rounded up the value to 10% as a first-order approximation to estimate the values of degree of folding for the folded state (> 0.9) and unfolded states (≤ 0.1). Anything in between is considered intermediate states. We plotted each of the three states as a function of FAD concentration (Fig. 2b).

Assuming that the fully folded protein is bound to FAD, we plotted the fraction of fully folded states, $f_F$, as a function of the log[FAD] (Fig. 2b), which was fitted to a single site binding isotherm curve to determine an apparent FAD dissociation constant: $f_F = a_F \frac{[FAD]}{K_{d,app} + [FAD]}$, where $a_F$ is the amplitude and $Kd_{app}$ is the apparent dissociation constant.

## Determining the number of clusters

We used the "mclust" package in R to perform model-based clustering of the FAD titration data and kinetic refolding data excluding zero values[39,85]. The datasets were considered separately as well as combined. We used the Bayesian Information Criterion (BIC)[86] to determine the number of clusters (Supplementary Fig. 4a, Supplementary Table 1). The criterion relies on the maximized log-likelihood function penalized for the complexity of the model to assess the fit of the mixture model at each possible number of clusters in a range of values. Mixture models that assume equal variance across all clusters (i.e., same variance parameter) and models that assume varying variance (i.e., different variance parameters for clusters), denoted respectively "E" and "V" models, were fit. The global analysis of the FAD titration and kinetic refolding datasets identified four distinct clusters with degrees of folding centered at $0.26 \pm 0.01$, $0.52 \pm 0.01$, $0.80 \pm 0.01$ and $1.0 \pm 0.01$ (mean ± standard error).

## Bootstrap analysis of clusters

We used bootstrapping to get standard errors for the estimated component parameters, including each cluster's mean and variance, as well as the cluster weights (i.e., the proportion of data in each cluster). Bootstrapping consists of repeatedly taking a random sample of size N with replacement from the data at hand and fitting the model to each bootstrap sample. The resulting bootstrap parameter estimates are then used to determine the standard errors for the respective parameter estimates. We considered a large number of bootstrap replications until the standard error estimates stabilized (this was achieved by around 10,000 replications). For example, Supplementary Fig. 4b shows the standard error for the estimated component mean for one of the clusters.

## Cluster analysis of cofactors contained in FAD

We investigated the effect of riboflavin, FMN, adenosine, AMP, ADP, ATP in the degree of folding of dCRY. Because of the differences in the chemical structures of these cofactors, the data from each moiety were clustered individually as opposed to the global clustering that was performed on the data with FAD. The BIC was used to determine the optimal number of clusters to group the data from each cofactor.

## Assignment of folded structures to each intermediate along the dCRY folding pathway

We sought to determine the secondary structures associated with each folding intermediate in dCRY that were identified through the clustering analysis. Based on the dCRY high-resolution structure (PDB: 4GU5)[80,87], we mapped the observed $\Delta Lc_T$ for clusters 2, 3 and 4 (with degree of folding of 0.26, 0.52, 0.80, respectively) using Eq. 2[27,88]:

$$\triangle Lc_T = (n + 1) \times L_{aa} - (X^{int}_{N \to m}) \quad (2)$$

$\Delta Lc_T$ describes the observed change in contour length of each folding intermediate (in nm). $n$ is the number of residues involved during the transition and $Lc$ is the contour length increment per amino acid (0.365 nm/aa). The second term, $X^{int}_{N \to m}$, is the distance between the N-terminus in the dCRY structure and the last residue (residue position "$m$") estimated for the folding intermediate. By moving the boundary of $m$, one residue position at the time, we obtained a family of WLC curves of $n$ residues and intermediate distances $X^{int}_{N \to m}$. The residue position giving the lowest root-mean-square-deviation

(R.M.S.D.) values from the WLC fit to the observed data are the optimal residues included in each folding intermediate (Supplementary Fig. 7). This approach considers that loops and random coils are unstable against mechanical force, and that secondary structures are stable when they are intact or fully folded[60,89].

## Multi-start evolutionary nonlinear OpTimizeR fitting models

Parameter optimization was carried out with the MATLAB (R2021b) toolbox MENOTR (Multi-start Evolutionary Nonlinear OpTimizeR)[72]. MENOTR is a hybrid genetic−NLLS algorithm that takes advantage of the respective strengths of both stochastic and deterministic parameter optimization algorithms while minimizing the weaknesses. MENOTR works by first generating a sampling of different values for each parameter in the chosen model. The sets of parameters are then subjected to different genetic operators such as mutation and crossover. The resultant sets of parameters are then ranked based on how well the parameters describe the experimental data. The best sets of parameters are then further optimized using NLLS methods. The resultant optimized parameters are then used in the genetic algorithm portion of the routine again. This process is continued in a cycle until there is no longer a significant difference in the starting and optimized parameters within an iteration. This toolbox provided a rigorous search of the error space compared to conventional NLLS methods.

MENOTR was used to examine three different models generated from the scheme shown in Fig. 3g. In the first case FAD only binds with the first intermediate, $I_1$. In the second case FAD binds with $I_2$ only. In the third case FAD binds to both $I_1$ and $I_2$. The set of ordinary differential equations describing each species within the scheme are shown below. The resulting optimized parameters are shown in Supplementary Table 2.

Ordinary Differential Equations for Fig. 3g:

$$\frac{d[U]}{dt} = -[U]k_1 + [I_1]k_{-1}$$

$$\frac{d[I_1]}{dt} = [U]k_1 - [I_1](k_{-1} + k_2) + [I_2]k_{-2} - [I_1][FAD]k_3 + [I_{1 \cdot FAD}]k_{-3}$$

$$\frac{d[I_2]}{dt} = [I_1]k_2 - [I_2]k_{-2} - [I_2][FAD]k_4 + [I_{2 \cdot FAD}]k_{-4}$$

$$\frac{d[I_{1 \cdot FAD}]}{dt} = [I_1][FAD]k_3 - [I_{1 \cdot FAD}](k_{-3} + k_5) + [I_{2 \cdot FAD}]k_{-5}$$

$$\frac{d[I_{2 \cdot FAD}]}{dt} = [I_{1 \cdot FAD}]k_5 - [I_{2 \cdot FAD}](k_{-5} + k_{-4} + k_6) + [I_{3 \cdot FAD}]k_{-6} + [I_2][FAD]k_4$$

$$\frac{d[I_{3 \cdot FAD}]}{dt} = [I_{2 \cdot FAD}]k_6 - [I_{3 \cdot FAD}](k_{-6} + k_7) + [F_{FAD}]k_{-7}$$

$$\frac{d[F_{FAD}]}{dt} = [I_{3 \cdot FAD}]k_7 - [F_{FAD}]k_{-7}$$

## Statistical analysis of refolding models

We used the $F$-test statistic to determine the best fitting model to the kinetic and FAD titration data shown in Fig. 3. First, for each dataset, we determined $f_{obs}$, for the pairwise comparison of the refolding models considered (Supplementary Table 3). In model 1, FAD binds to both intermediates, $I_1$ and $I_2$. In model 2, FAD only binds to $I_1$. In model 3,

FAD only binds to $I_2$. The $F$-test statistic is calculated as follows (Eq. 3):

$$f_{obs} = \left(\frac{\chi_j^2}{\nu_j}\right) \bigg/ \left(\frac{\chi_k^2}{\nu_k}\right) \tag{3}$$

where $\chi^2$ is the Chi square statistics for the fitted model and $\nu$ corresponds to its degrees of freedom. The indexes $j$ and $k$ refer respectively to the smaller and larger models being compared. The values for $\chi^2$, $\nu$ and $f_{obs}$ are listed in Supplementary Table 3. The test statistic, $f_{obs}$, follows an $F$-distribution with degrees of freedom corresponding to the models being compared. This allows us to evaluate a p value assessing if the smaller model fits the data as well as the larger model. The results are listed in Supplementary Table 3. These tests were done in *Mathematica* 12 (Wolfram Research, Inc.).

### Calculation of average first-passage time from the unfolded to the native state

We determined the average time dCRY takes to fold into the native, FAD-bound state starting from the unfolded state. Briefly, we rewrote the net rate constants from the unfolded to the folded state as the original rate constants in Fig. 3g multiplied by the partition between forward and backward flux (Supplementary Fig. 5a)[90]. The net rates in the scheme in Supplementary Fig. 5a are defined as (Eqs. 4–8):

$$k_7' = k_7 \tag{4}$$

$$k_6' = k_6 \frac{k_7'}{k_{-6} + k_7'} \tag{5}$$

$$k_4' = (k_4[FAD]) \frac{k_6'}{k_{-4} + k_6'} \tag{6}$$

$$k_2' = k_2 \frac{k_4'}{k_{-2} + k_4'} \tag{7}$$

$$k_1' = k_1 \frac{k_2'}{k_{-1} + k_2'} \tag{8}$$

The values of the rate constants were obtained from Supplementary Table 2 (Global Analysis of Kinetic Refolding and FAD Titration Data). Using the net rates, the average first-passage time from U to $F_{FAD}$ as a function of [FAD] is (Eq. 9):

$$\tau_{U \to F_{FAD}} = \frac{1}{k_1'} + \frac{1}{k_2'} + \frac{1}{k_4'} + \frac{1}{k_6'} + \frac{1}{k_7'} \tag{9}$$

The plot of $\tau_{U \to F_{FAD}}$ vs [FAD] is shown in Supplementary Fig. 5b. When [FAD] = 10 nM, the average folding time is 30 s. The cumulative probability distribution of an exponential function using a time constant of 30 s indicates that the probability of folding at 40 s is 0.73, a value that is consistent with the experimentally obtained folded fraction of 0.66 ± 0.4 (Fig. 3i).

### Reporting summary
Further information on research design is available in the Nature Portfolio Reporting Summary linked to this article.

## Data availability
The data generated in this study, including all figures and tables, are available on request. The published dCRY high-resolution structure was obtained from the Protein Data Bank with the accession code 4GU5. Source data are provided with this paper.

## Code availability
The code for data clustering with bootstrapping to generate Figure panels 2b, 3d, e, i, 4b–d, and Supplementary Tables 1, 4 is freely available at https://doi.org/10.5281/zenodo.7600414. The code for global fitting shown in Fig. panels 3d, e, i and Supplementary Table 2 is freely available at https://doi.org/10.5281/zenodo.7600386.

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

## Acknowledgements

The authors wish to acknowledge Joanne Widom from the Crane lab for cloning and expression of the original tagging enzymes and modified CRY. We thank Shixin Liu from The Rockefeller University for assistance in the calculation of average first-passage times. This work was supported by NSF grants MCB1715572 (to R.A.M.), MCB-1412624 and MCB-1817749 (to A.L.L.), and NIH grants 1R15GM135866 (to R.A.M.) and R35GM122535 (to B.R.C). L.L.G. gratefully acknowledges support from the Henry Luce Foundation's Clare Boothe Luce Scholarship and from the Barry Goldwater Scholarship.

## Author contributions

S.F. designed, conducted, and analyzed the research, and wrote the manuscript. L.G., Z.I. and M.G.T. analyzed the data. C.F. conducted research and edited the manuscript. A.L.L. analyzed the research. B.R.C. analyzed the research and wrote the manuscript. R.A.M. designed and analyzed the research and wrote the paper.

## Competing interests

The authors declare no competing interests.
