## [Peer Review File · Nature Communications]

REVIEWER COMMENTS

Reviewer #1 (Remarks to the Author):

Foroutannejad et al. undertook the formidable task of enhancing our knowledge about the folding of a 542-residue protein that, to make matters even more complicated, contains a cofactor. Use was made of optical tweezers to study the light sensing *Drosophila* cryptochrome (dCRY), which uses FAD for its function. The authors suggest that the folding of dCRY involves various steps, starting with parts of the protein that fold first and independently of FAD, followed by cofactor binding and co-folding around the flavin.

The brave and challenging work of Foroutannejad et al. raises the following questions and comments.

* Title: Remove 'energy landscape', as the manuscript does not determine the energy landscape of the protein under investigation.

* Abstract, line 16: remove 'energy landscape', as the manuscript does not determine the energy landscape of the protein under investigation.

* Abstract, line 23: The word 'Surprisingly' should be removed, as it is amply demonstrated that proteins can start folding without the presence of a cofactor.

* Abstract, line 29: change '... (i.e., AMP and ADP) only allow partially folded structures' into '(i.e., AMP and ADP) only allow formation of partially folded protein structures'.

* In the introduction, first paragraph, line 43 mention is made that upon incorporating cofactors the thermodynamic properties of a protein can alter. The authors should consider incorporating the following reference, as it experimentally reveals at the amino acid residue-level the thermodynamic effects of the incorporation of FMN into a flavoprotein (Bollen, Y. J., Westphal, A. H., Lindhoud, S., van Berkel, W. J. H. & van Mierlo, C. P. M. (2012) *Nat. Commun.* 3:1010, 9 pages, DOI: 10.1038/ncomms2010; "Distant residues mediate picomolar-binding affinity of a protein cofactor").

* In the introduction, second paragraph, mention is made of flavoproteins that fold into their native state independently of the cofactor (references 11-13). I judge that the following publication should be added, as it demonstrates that during folding of *Azotobacter vinelandii* flavodoxin, FMN only binds to native apoflavodoxin (Bollen, Y. J. M., Nabuurs, S. M., van Berkel, W. J. H. & van Mierlo, C. P. M. (2005) *J. Biol. Chem.* 280 (9), 7836-7844, "Last in, first out: the role of cofactor binding in flavodoxin folding").

* Page 3, lines 80-81; change 'In contrast, the adenosine ring of FAD with one or two phosphate groups (i.e., AMP or ADP) only forms partially folded structures' into 'In contrast, the adenosine ring of FAD with one or two phosphate groups (i.e., AMP or ADP) causes formation of only partially folded protein structures'.

* Page 4, line 97: the reference to supporting figure S2 is not appropriate here.

* Page 4, line 126: the authors mention testing of the folding of the protein to the native state in the absence of the cofactor. In the Supplementary Information it is stated, on p. 25, line 61, that the protein was dialyzed overnight in dCRY buffer at 4 C. Did the authors verify, via for example FAD absorption, that the protein of interest is indeed in the apo form?

* On p. 5, line 154, the authors refer to the apparent FAD dissociation constant they determined. Is the FAD dissociation constant for dCRY known from other experiments and how does this value relate to the author's determination?

* On page 6, a scheme as model for the folding of dCRY is shown. Global fitting of this scheme to the data leads to fourteen rate constants. Is it possible that the data are over fitted?

* On p. 6, lines 211 and 212, mention is made about microscopic (i.e., state specific) equilibrium dissociation constants (k_{-3}/k_3) and (k_{-4}/k_4) of 0.29 nM and 0.15 nM., respectively What is meant by 'state specific'? This would imply that at $[FAD] = 25$ micromolar (Fig. 1c) the stability of intermediate I1-FAD is about 6.7 kcal/mol and of intermediate I2-FAD is about 7.1 kcal/mol (considering that $\Delta G = -RT \ln(K)$)? Compared to the stabilities of the other intermediates, as calculated from the microscopic rate constants of Table S2, the binding of FAD to either intermediates I1 or I2 would then be the dominant contribution to the stability of the final FAD-containing properly folded protein.

* Page 7, line 215, the value 5.1 should be 4.7 according to my calculation.

* On page 7, line 238, mention is made of Mg²⁺ interacting with FAD in the protein structure. However, in the dCRY buffer (p. 12, line 470) no Mg²⁺ is present. Was Mg²⁺ present in the buffer of the experiments that led to Figures 1 and 3? If not, why?

* On page 10, line 354, the authors mention that the association of the C-terminal tail (CTT) to the C-terminal coupled motif (CCM) is probably the least stable folding step of dCRY. However, the values for k₁ and k₋₁ lead to a deltaG of -0.1 kcal/mol for intermediate I1 versus U. The values for k₇ and k₋₇ (line 220, page 7) lead to a deltaG of -0.8 kcal/mol of FFAD versus I3-FAD. Consequently, the association of the C-terminal tail (CTT) to the C-terminal coupled motif (CCM) is not the least stable folding step of dCRY.

* On page 11, line 413, the authors mention 'Surprisingly, we find that FAD is strictly required for dCRY to attain its native state, i.e., without FAD the protein never folds to its biologically functional structure.' The word 'surprisingly' should be removed, as the biologically functional structure of dCRY obviously needs FAD.

Reviewer #2 (Remarks to the Author):

The cofactor-dependent folding energy landscape of a light-sensing protein revealed by single-molecule pulling experiments

The authors present a systematic study of the FAD-dependent folding energy landscape of *Drosophila* cryptochrome (dCRY), a light-sensing protein. An optical tweezer-based single-molecule manipulation technique explores the intermediate states along the FAD-dependent folding pathway to the native folded state of the protein. The study dissects which intermediates are independent of the FAD-binding and forms scaffolds for further cofactor binding to achieve the biologically significant conformation. The results obtained through single-molecule experiments agree quantitatively with previously recorded results from structural analysis. The experiment is rigorous and gives insight into the molecular mechanism of the underlying process of the protein.

Conceptually, the work is well designed and executed with data analyzed by new methods. I think this contribution is of significant interest to justify publication after a revision to address the following concerns:

General comments:

1. The states reported in this work are observed by a pulling speed 75 nm/s. How does the pulling speed affect the state populations and the derived transition rates?
2. Since the object under study is a light-sensing protein, can authors comment on the role of intermediate states (e.g., for states with a degree of folding close to 1) in signaling? To be more specific, at which intermediate state does the protein lose its function?
3. Does the intense trapping light affect dCRY structure and folding?

Minor comments:

1. Figure 1c and e: Should the x-axis be “extension” instead of “trap position”?
2. Line 154-155: may add a reference to the SI at the end of the sentence for the fitting equation.
3. The x-axis label in Figure 3d-f denotes the waiting time for the refolding of the protein (i.e., refolding time). I suggest explicitly mentioning “refolding” in the axis label here (up to the authors).
4. Can authors define quantities such as density and fraction of states used in Figure 3?
5. Line 180: “The analysis does not indicate which cluster incorporates FAD-bound and unbound dCRY states.” However, the section starting at Line 315 and Figure 5 do indicate that these intermediates can be mapped based on the cluster analysis. The authors may want to clarify this in line 180 or add a comment referring to the later section.
6. Line 190 in SI: in the second term inside the parenthesis, $k_{-2} \rightleftharpoons k_{2}$
7. Lines 194 and 195 in SI: $[I_{\{F.FAD\}}]$ should be $[FAD]$ following scheme 1.
8. To analyze the extension measurements from optical tweezers, the authors used crystal structural information of dCRY. This methodology to use protein atomic structures to interpret the extension data from optical tweezers is previously published (Reban, AA et al, BJ, 110: 441, 2016), which may be cited.

Reviewer #3 (Remarks to the Author):

Through the help of optical tweezers and in-depth kinetic analysis, the present single-molecule study captured the unfolding-folding of the *Drosophila* cryptochrome (dCRY) with the help of the co-factor flavin adenosine diphosphate (FAD) and attempted to identify the mechanism for the same. Five distinct states were identified through which a kinetic scheme was developed and using statistical techniques, the kinetic rate constants were determined. The results and explanations are well-presented. Overall, the study provides important insights about the absolute need of FAD in allowing dCRY to attain a

stable native state via experimentally observed distinct intermediate states supported by quantitative estimation of kinetic rate constants. The work maybe further refined after addressing some points.

1) Line 126: "Figure 1e" is mistyped as "Figure 1c".

2) Line 154: The details for the equation used to determine the apparent FAD dissociation constant should be provided.

3) From the discussion in the section "Contribution of FAD Moieties to dCRY Folding", it is evident that isoalloxazine ring as well as the adenosine moiety are individually able to aid in complete folding of dCRY. What is the relative efficiency of each of them in promoting dCRY folding, especially with respect to their interactions with the protein? Do both these moieties follow the same pathway?

4) Do different FAD moieties lead to separate folding mechanisms or is Scheme 1 universally followed?

5) A comparative study between FAD binding to clusters I 1 and I 2 should provide significant insights about the mechanism of the folding process and could be explored.

6) The color scheme for figure 3, the colors used for C3 and C4 are almost indistinguishable.

7) Did the authors determine the time for the protein to attain the native structure from the unfolded state? A statistical analysis of the folding time could provide insights to the stochasticity of the unfolding-folding process.

8) In Supporting Fig. S4 (a-b), please define "E" and "V" mentioned in the respective legends.

Reviewer #4 (Remarks to the Author):

Foroutannejad et al have made an important experimental contribution towards understanding protein folding for large proteins with co-factors. It was very interesting to read.

The strength of the study is, in my opinion, the description of the folding process of dCRY and how it involves several steps with co-factor involvement. I found the data in figure 1c and 1d very impressive. And I like the authors attempt of describing these steps in the unfolding process, albeit I have some concerns (below). Another strength of the paper is the systematic investigation of different moieties of the cofactor, which I think was a great and thorough piece of work, and aid in a better understanding of the folding process of dCRY, as well as the role of the components of the co-factor.

I have some concerns regarding the clustering method:

- I am in doubt how much to trust the clustering algorithm. It would be great to see it tested against a known system (with real experimental or synthetical data), to benchmark if the method can reproduce the correct (known) number of clusters. This is probably a lot of work, but if such test/benchmarking cannot be presented, then I think the authors should be slightly more humble about their conclusions, number of clusters etc, and discuss potential limitation or pitfalls of the method.

- The authors claim to use BIC and ICL. But these are two different model assessment equations (see e.g. <https://arxiv.org/pdf/1411.4257.pdf>), which do not necessarily result in the same number of clusters. Maybe the methods gave the same result for this dCRY (although, from figure SI-3 it is not clear that $n=4$ is a minimum or a maximum of ICL). The authors definitely have to describe this part more thoroughly and clearly. Why use both BIC and ICL? what if they give different results in the clustering - which one to choose then, and why?

- A small, but general comment, is that bootstrapping is highlighted on several places in the manuscript, e.g., in abstract: "that integrates clustering, bootstrapping, and global fitting...". Bootstrapping is "just" a way to determine the errors, if I understand the presented method correctly, and should therefore, in my opinion, not be highlighted so much. It could just be mentioned in the method section.

Following is a list of suggestions for improvements. Major points must be addressed before publication. These are marked with an asterisk "*"

title:

"a light-sensing protein" is a bit unspecific

line 15-17: it is an unpleasantly long starting sentence - any book on scientific writing will tell you that is a bad idea.

line 16:

I suggest you remove "the" in "the polypeptide", unless you refer to a specific polypeptide

line 16

why do you write "polypeptide" here, when you use the term "protein" or "multidomain protein" in the rest of the paper? Personally, I think "protein" is a better term.

line 20: "that binds FAD, one of the most common, complex cofactors" could be changed to: "that binds the cofactor FAD"

Alternatively, please specify what you mean by "common" and "complex"

line 23: "diffusion-limit" in two words: "diffusion limit"

line 26: "chemical" - isn't that word redundant?

* line 30: "with structural data" - be more specific. what structural data?

* line 32-34: generally useable - examples of other systems?

line 37-43: are refs 1-7 necessary?

* line 44: what is meant by "bulk studies"?

line 51-52: it would be interesting with a reference or argument for why FAD is "common" as well as a specification of what is meant by "large" in this context

* line 53-54: what is meant by "in bulk"? do you mean in solution?

line 54: does "it" here refer to single molecule techniques? then it should be plural

line 56: "importantly". In my opinion, the other points are equally important.

line 52-57: this is a very long sentence, which makes it hard to read. Could you rephrase, or split in two?

line 58-72: some abbreviations could be avoided: CTT, CCM - are they necessary?

*Figure 1a: the structure in figure 1a (left) is too small, and it is difficult to link what is described in the text with the figure. E.g., it would be easier to follow, if the three loops were annotated on the structure. And it is hard to see the 4-helix bundle. This is a pity, as it is an important figure panel in order to understand the paper.

*line 82: "structural data". Please, be more specific

Figure 1b: no hyphen in Avi-tag and ybbR-tag. Annotations "CoA" and "B" are written with too small font. Why not write out "Biotin"?

abbreviation SA and AD not explained.

line 96: please, write out "base pair" instead of abbreviating

line 97: a bit unclear what the 5' end of a double helix is... there is a 5' and a 3' in both ends.

line 100: what specifically do the authors refer to in SI here?

*line 100-102: the spectroscopic data show that dCRY is functional, but not necessarily that the fold is unchanged.

*Corrections to SI fig S2:

SI, fig S2a: axis labels needed.

SI, fig S2bc: what does A403 and A365 refer to?

why negative labels on a y-axis showing percentage? should it be 100% instead of 1% as max? in that case, the max y-value should be 100, not 105.

(and plotting in Excel - hmm... If everything was notated correctly etc, I wouldn't complain. But I can't help feeling the figure is a bit unfinished - not ready for a thesis, and definitely not for a peer-reviewed journal publication).

line 120: when uncertainty is 15, you should not give the mean with 4 significant figures. Just write 206 +/- 15. https://en.wikipedia.org/wiki/Significant_figures

*line 120: just to be sure. Do you mean standard error = standard deviation of the mean. Or, the standard deviation for the N experiments? You should give the standard error as uncertainty on the mean.

line 122-123: please write the equation on its own line, for clarity

line 123: I think it is optimistic to estimate this number with an accuracy that can justify 4 significant figures. Probably writing 198 nm is fine.

* line 126: figure 1c is not "in absence of the cofactor", but has a 25 μ M conc of FAD. Do you mean fig 1e? please correct or clarify.

figure 1c-f: colors are inconsistent, if I read the figure correctly. left side: blue is unfolding, red is refolding. But right side: blue is with FAD, red is without FAD.

line 143: what is referred to in SI here?

* line 140: you write that the degree of folding is between 0 and 1, but clearly (from the figure), it can exceed 1. I do see the logic after reading a couple of times, but it should be more clearly written.

line 150: In my opinion, there is no need to refer to the figure again, you are in the middle of a discussion of it.

*line 154: it is not a dissociation constant. calling it so is confusing - indicating that you have made a binding experiment.

line 150-152: that whole discussion of intermediates seems a bit redundant. The message is clear-cut without.

* figure 2: very nice data. But why do you want to do that division into (arbitrarily chosen) subdivisions? A plot of average degree of fold as function of concentration would summarise the findings without subdivisions, and in a more simple and reproducible way, instead of Fig 2b. Please change this, or justify your choices. Also, discuss the subdivisions. Why these? Would the results change if you chose other subdivisions? Would this subdivision work for other systems?

fig 2a: please use the same ylimits for all conc - that would make it much easier to compare.

*fig 2b: something is wrong with the x-axis. $\log(0)$ is mathematically undefined, but you have a 0 M concentration. Where is that plotted? when you take the log of a conc, the units are not longer M, but $\log(M)$

line 154-155: why give uncertainty as 95% CI (2 std dev, assuming normal distr), when you gave results with 1 sd uncertainty previously (and later) in the manuscript. It is not wrong, but a bit inconsistent.

line 158: the multiple steps in the unfolding are, in my opinion, more clear from fig 1c than from fig 2a.

figure SI-3: please use the same ylimits so histograms can be compared visually. How come you see any folding in 0 s? this should be explained more thoroughly

*line 181-182: "direct" is a bit of an overstatement. It is a rather indirect method. The data in figure 1c are, on the other hand, direct evidence of unfolding steps.

*line 198-201: please provide p-values from the F-test in the main text. Please provide N, and describe what you have used as number of degrees of freedom.

figure 4: please denote riboflavin, ribitol and FMN on figure 4a.

SI-Fig 4: unfolded stated -> unfolded states

this sentence is difficult to understand: "c- the standard error on the fractional populations was determined by the number of bootstrap iterations performed was increased"

SI-Fig 4c: standard error has units - please provide these

*SI-Fig 4: please write what "E" and "V" means

Dear Reviewers:

We thank you for taking the time to carefully read our manuscript, and provide important suggestions, comments and critiques. The comments we received guided us to better explain our approaches, clarify concepts and connect our findings with previously published studies. Below you will find detailed responses to all major and minor comments.

We hope that we were able to address your questions in a clear manner. All of the new additions and modifications in the manuscript are in blue font. The response to each specific question is also in blue font. We believe that the revised manuscript provides a clearer presentation of our data and analysis.

Reviewer #1 (Remarks to the Author):

Foroutannejad et al. undertook the formidable task of enhancing our knowledge about the folding of a 542-residue protein that, to make matters even more complicated, contains a cofactor. Use was made of optical tweezers to study the light sensing *Drosophila* cryptochrome (dCRY), which uses FAD for its function. The authors suggest that the folding of dCRY involves various steps, starting with parts of the protein that fold first and independently of FAD, followed by cofactor binding and co-folding around the flavin.

The brave and challenging work of Foroutannejad et al. raises the following questions and comments.

* Title: Remove 'energy landscape', as the manuscript does not determine the energy landscape of the protein under investigation.

We agree with the reviewer and have replaced the words 'energy landscape' in the title with 'mechanism.'

* Abstract, line 16: remove 'energy landscape', as the manuscript does not determine the energy landscape of the protein under investigation.

Similarly, we have replaced the words 'reshape the folding energy landscape' with 'modulate the folding mechanism.'

* Abstract, line 23: The word 'Surprisingly' should be removed, as it is amply demonstrated that proteins can start folding without the presence of a cofactor.

As suggested by the reviewer, we deleted the word "Surprisingly.'

* Abstract, line 29: change ‘... (i.e., AMP and ADP) only allow partially folded structures’ into ‘(i.e., AMP and ADP) only allow formation of partially folded protein structures’.

We thank the reviewer for this suggestion and we added the words ‘formation of’ to the Abstract.

* In the introduction, first paragraph, line 43 mention is made that upon incorporating cofactors the thermodynamic properties of a protein can alter. The authors should consider incorporating the following reference, as it experimentally reveals at the amino acid residue-level the thermodynamic effects of the incorporation of FMN into a flavoprotein (Bollen, Y. J., Westphal, A. H., Lindhoud, S., van Berkel, W. J. H. & van Mierlo, C. P. M. (2012) Nat. Commun. 3:1010, 9 pages, DOI: 10.1038/ncomms2010; “Distant residues mediate picomolar-binding affinity of a protein cofactor”).

We thank the reviewer for bringing to our attention the references above-mentioned, which we added at the of paragraph 1 of Introduction.

* In the introduction, second paragraph, mention is made of flavoproteins that fold into their native state independently of the cofactor (references 11-13). I judge that the following publication should be added, as it demonstrates that during folding of *Azotobacter vinelandii* flavodoxin, FMN only binds to native apoflavodoxin (Bollen, Y. J. M., Nabuurs, S. M., van Berkel, W. J. H. & van Mierlo, C. P. M. (2005) J. Biol. Chem. 280 (9), 7836-7844, “Last in, first out: the role of cofactor binding in flavodoxin folding”).

We agree with the reviewer that the suggested reference belongs with references 11-13 in the original manuscript.

* Page 3, lines 80-81; change ‘In contrast, the adenosine ring of FAD with one or two phosphate groups (i.e., AMP or ADP) only forms partially folded structures’ into ‘In contrast, the adenosine ring of FAD with one or two phosphate groups (i.e., AMP or ADP) causes formation of only partially folded protein structures’.

As suggested by the reviewer, we made the change and replaced ‘only forms’ with ‘causes formation of.’

* Page 4, line 97: the reference to supporting figure S2 is not appropriate here.

The reviewer is correct, and we have removed the reference to supporting figure S2 in the text.

* Page 4, line 126: the authors mention testing of the folding of the protein to the native state in the absence of the cofactor. In the Supplementary Information it is stated, on p. 25, line 61, that the protein was dialyzed overnight in dCRY buffer at 4 C. Did the authors verify, via for example FAD absorption, that the protein of interest is indeed in the apo form?

We measured the presence of FAD in the sample after dialysis using absorption spectroscopy to confirm the absence of the cofactor. We have included this spectrum as new Supporting Figure 2 (new panel 'd'). We also modified the main text in the section entitled Online Materials and Methods.

* On p. 5, line 154, the authors refer to the apparent FAD dissociation constant they determined. Is the FAD dissociation constant for dCRY known from other experiments and how does this value relate to the author's determination?

To the best of our knowledge, a quantitative measurement of the K_d for FAD and dCRY has not been reported. The reason behind this omission is the challenge of working with dCRY in bulk in the apo state, as it is a very aggregation-prone protein. This is one of the strengths of our single molecule approach with optical tweezers, namely, by using extremely dilute protein conditions and sampling one protein molecule at the time, we prevent aggregation and therefore interrogate the change in states (unbound vs. bound) as a function FAD concentration.

* On page 6, a scheme as model for the folding of dCRY is shown. Global fitting of this scheme to the data leads to fourteen rate constants. Is it possible that the data are over fitted?

We placed great attention to identify the simplest model that provided the best fitting statistics. We tried using simpler models with a smaller number of fitted parameters than the one shown in Scheme 1. In the resubmitted version of this manuscript, we included the statistical test to determine the smaller refolding model that fits the data well. The result of the statistical analysis is summarized in a new Supporting Table S3 and explained in detail at the end of the Supplementary Information. We also modified the main text and reference Supporting Table S3 at the end of the 3rd paragraph under the subtitle "A Complex dCRY Folding Pathway is Coupled to FAD Binding."

* On p. 6, lines 211 and 212, mention is made about microscopic (i.e., state specific) equilibrium dissociation constants (k_{-3}/k_3) and (k_{-4}/k_4) of 0.29 nM and 0.15 nM., respectively. What is meant by 'state specific'? This would imply that at [FAD] = 25 micromolar (Fig. 1c) the stability of intermediate I1-FAD is about 6.7 kcal/mol and of intermediate I2-FAD is about 7.1 kcal/mol (considering that $\Delta G = -RT\ln(K)$)? Compared to the stabilities of the other intermediates, as calculated from the microscopic rate constants of Table S2, the binding of FAD to either intermediates I1 or I2 would then be the dominant contribution to the stability of the final FAD-containing properly folded protein.

The reviewer brought up an important point that we did not mention in the manuscript. It is correct that the steps in which FAD binds contribute the most to the stability of the final FAD-bound protein. This point motivated us to investigate the literature for studies that have determined the concentration of FAD in eukaryotic cells. A recent study by Hühner and collaborators (reference at the end of this response) determined that the concentration of FAD in mammalian cells ranges between 8-240 nM. In *E. coli*, the concentration is much higher, reaching 170 μ M (Bennet et al., 2009, reference at the end of this answer). We were unable to find concentrations of FAD in *Drosophila* cells. Nonetheless, we used the values of eukaryotic cells to determine a range of stabilities for I1-FAD and I2-FAD. Using $\Delta G = -RT\ln(K)$, the stability for I1-FAD range is 2-4 kcal/mol, and for I2-FAD is 2.2-4.4 kcal/mol. These values are higher than the estimated stability of other states like I1 (vs. U), I2 (vs. I1), I2-FAD (vs. I1-FAD), I3-FAD (vs. I2-FAD) and F-FAD (vs. I3-FAD). Therefore, at the end of the last paragraph of the section entitled "A Complex dCRY Folding Pathway is Coupled to FAD Binding" we added a line referring to the point brought up by the reviewer indicating that I1-FAD and I2-FAD are the intermediates that contribute the most to the stability of fully folded FAD-bound protein.

Regarding the terminology 'state specific,' we were implying that the equilibrium constants determined correspond to microscopic states, and not macrostates. Clearly, we generate more confusion than anything, and we have now omitted the words 'state specific' in the revised version of the manuscript.

- Hühner J, Ingles-Prieto Á, Neusüß C, Lämmerhofer M, Janovjak H. Quantification of riboflavin, flavin mononucleotide, and flavin adenine dinucleotide in mammalian model cells by CE with LED-induced fluorescence detection. *Electrophoresis*. 2015 Feb;36(4):518-25. doi: 10.1002/elps.201400451. Epub 2015 Jan 22. PMID: 25488801.
- Bennett, B., Kimball, E., Gao, M. et al. Absolute metabolite concentrations and implied enzyme active site occupancy in *Escherichia coli*. *Nat Chem Biol* 5, 593–599 (2009). <https://doi.org/10.1038/nchembio.186>

* Page 7, line 215, the value 5.1 should be 4.7 according to my calculation.

We thank the reviewer for catching that error. We have changed the value to 4.7.

* On page 7, line 238, mention is made of Mg²⁺ interacting with FAD in the protein structure. However, in the dCRY buffer (p. 12, line 470) no Mg²⁺ is present. Was Mg²⁺ present in the buffer of the experiments that led to Figures 1 and 3? If not, why?

The main goal for this study was to use a bottom-up approach to dissect the role of FAD and its moieties as well as Mg²⁺ in the mechanism of folding of dCRY. Therefore, we first dissected the effect of FAD (Figures 1 and 3) and its moieties (Figure 4b-c) before probing the role of Mg²⁺ (Figure 4d and Supporting Figure S5). We find that the presence of Mg²⁺ (0.5mM) has little to no effect on the distributions of species (FAD=25uM, refolding time 20 s). While Figure 1 shows data with no Mg²⁺, it is indistinguishable to the data with Mg²⁺. To clarify this point, in this revised manuscript we added to the legend of Figure 1 that indistinguishable results were obtained in the presence of Mg²⁺. Moreover, given that the distributions of species (intermediates and fully folded states) were the same when using FAD with or without Mg²⁺, we surmise that the rate constants dissected for Scheme 1 may not be significantly altered by the presence of the cation. We cannot discard that some changes may be observed when using other cofactors like ADP and ATP, which showed significant differences in species distributions between presence and absence of Mg²⁺. Nonetheless, we think that such work corresponds to a follow-up study.

* On page 10, line 354, the authors mention that the association of the C-terminal tail (CTT) to the C-terminal coupled motif (CCM) is probably the least stable folding step of dCRY. However, the values for k₁ and k₋₁ lead to a deltaG of -0.1 kcal/mol for intermediate I1 versus U. The values for k₇ and k₋₇ (line 220, page 7) lead to a deltaG of -0.8 kcal/mol of FFAD versus I3-FAD. Consequently, the association of the C-terminal tail (CTT) to the C-terminal coupled motif (CCM) is not the least stable folding step of dCRY.

We thank the reviewer for pointing this out. Indeed, formation of unbound intermediates like I1 is less stable than folding of the CTT to the protein core. We therefore have changed the text and deleted “the least stable folding step” and replaced it with “the last folding step,” which reflect the results from the global analysis of the kinetic data.

* On page 11, line 413, the authors mention ‘Surprisingly, we find that FAD is strictly required for dCRY to attain its native state, i.e., without FAD the protein never folds to its biologically functional structure.’ The word ‘surprisingly’ should be removed, as the biologically functional structure of dCRY obviously needs FAD.

The reviewer makes a valid point in that biologically active dCRY requires FAD. We have removed the word “Surprisingly” from the text, but we have added that other structurally homologous dCRYs, for example the mammalian CRY, folds and functions without the cofactor.

Reviewer #2 (Remarks to the Author):

The cofactor-dependent folding energy landscape of a light-sensing protein revealed by single-molecule pulling experiments

The authors present a systematic study of the FAD-dependent folding energy landscape of *Drosophila* cryptochrome (dCRY), a light-sensing protein. An optical tweezer-based single-molecule manipulation technique explores the intermediate states along the FAD-dependent folding pathway to the native folded state of the protein. The study dissects which intermediates are independent of the FAD-binding and forms scaffolds for further cofactor binding to achieve the biologically significant conformation. The results obtained through single-molecule experiments agree quantitatively with previously recorded results from structural analysis. The experiment is rigorous and gives insight into the molecular mechanism of the underlying process of the protein. Conceptually, the work is well designed and executed with data analyzed by new methods. I think this contribution is of significant interest to justify publication after a revision to address the following concerns:

General comments:

1. The states reported in this work are observed by a pulling speed 75 nm/s. How does the pulling speed affect the state populations and the derived transition rates?

The reviewer makes two important points: (1) the effect of pulling speed on state populations and (2) the effect of pulling speed on the derived transition rates.

We will first discuss the effect of pulling speed on state populations. A pronounced effect of the pulling speed in mechanical unfolding experiments is the force at which a state unfolds. In general, the higher the pulling speed, the higher the force at which a state will unfold (Equation 18 in Bustamante et al. 2004; Equation 6 in Dudko, 2016. References at the end of this question). However, there are other factors that modulate the extent to which the pulling speed affects the observed unfolding force. For instance, if the distance to the transition state is large, the effect of pulling speed over the unfolding force is actually quite minimal, even if the speed is increased substantially (i.e., 10 times faster). Altogether, this descriptive explanation indicates that the pulling speed may have a pronounced effect on the force at which a particular state is observed but not at its probability of being sampled. This is because in our experiments, the protein is allowed to sample various states along its folding pathway at very low forces (1 pN), where we aim to minimize the effect of force on the rates at which these states interconvert. When we pull on the protein, we focus on the total change in extension rather than its unfolding force in order to determine the degree to which the protein refolded given a specific set of experimental conditions.

The second point the reviewer makes is the effect of force on the kinetic rates observed. It is possible that if we allow the protein to refold and rebind the cofactor at a higher force than what we used (i.e., using 3 pN instead of what we use: 1 pN) the kinetic parameters

may change, likely resulting in slower rates of refolding. Therefore, to minimize force effects on the observed rates, we used the lowest force possible in our instrument – one that approaches zero pN. We believe that obtaining force-dependent rate constants will be informative, yet beyond this initial study because of the significant additional set of experiments that such an endeavor would require. To clarify the point made by the reviewer, we added at the end of the second paragraph of **Conclusions** the following sentence: “We note that the fitted parameters obtained in this study were obtained at 1 pN. While this force is close to zero pN, future experiments in which the refolding force is varied will be required to identify which kinetic rate constants may have a strong force dependence.”

- Bustamante C, Chemla YR, Forde NR, Izhaky D. Mechanical processes in biochemistry. *Annu Rev Biochem.* 2004;73:705-48. doi: 10.1146/annurev.biochem.72.121801.161542. PMID: 15189157.
- Dudko OK. Decoding the mechanical fingerprints of biomolecules. *Q Rev Biophys.* 2016 Jan;49:e3. doi: 10.1017/S0033583515000220. Epub 2015 Oct 26. PMID: 26498560.

2. Since the object under study is a light-sensing protein, can authors comment on the role of intermediate states (e.g., for states with a degree of folding close to 1) in signaling? To be more specific, at which intermediate state does the protein lose its function?

This is another important question, and based on our single molecule studies and previous results from other, we believe dCRY is not functional from the unfolded states until the second intermediate bound to FAD ($I_2 \cdot \text{FAD}$). The formation of $I_3 \cdot \text{FAD}$ provides all the scaffolding for the protein oscillate between inactive and active states. Based on functional studies, the inactive state (dark state) is the fully folded state, $F \cdot \text{FAD}$, wherein the C-terminal tail and C-terminal linker are folded into the C-terminal coupled motif (CCM) and FAD. The active state (light activated state) would correspond to $I_3 \cdot \text{FAD}$ wherein the CTT and C-terminal linker have not docked into the CCM.

We added at the end of the 3rd paragraph of section “**Mapping Folding and FAD-Bound Intermediates Along the dCRY Folding Pathway**” the following sentence in order to emphasize the point made by the reviewer: Altogether, our single molecule studies indicate that $I_3 \cdot \text{FAD}$ is likely the first functional intermediate of dCRY, i.e., intermediates prior to $I_3 \cdot \text{FAD}$ do not have the scaffolding to allow changes in conformation dependent on the FAD redox state.

3. Does the intense trapping light affect dCRY structure and folding?

It has been shown by us and others that optical tweezers folding studies with proteins reveal fully folded states that are in agreement with high resolution structures. This indicates that the laser light doesn't affect the protein fold. This is in part because the laser light wavelength used in many of these studies, including the present one, is near 800 nm, which doesn't heat up water or generate reactive oxygen species. In our study, dCRY also folds to the native state (based on the crystal structure) when FAD is present. Moreover, to further mitigate other potential problems due to the laser light, we immobilized the protein on the bead that is held by the micropipette tip, instead of the bead that is held in the optical trap (Figure 1A).

Minor comments:

1. Figure 1c and e: Should the x-axis be "extension" instead of "trap position"?

We thank the reviewer for pointing this out. Indeed, the raw data of our instrument corresponds to Trap Position in nm. See reference below:

- Smith SB, Cui Y, Bustamante C. Optical-trap force transducer that operates by direct measurement of light momentum. *Methods Enzymol.* 2003;361:134-62. doi: 10.1016/s0076-6879(03)61009-8. PMID: 12624910.

Fortunately, the analysis of this work that is focused on total changes in extension upon unfolding does not require us to determine the molecular end-to-end distance between the two beads. This is because the changes in molecular extension at the same force cancel the contributions due to trap compliance or changes in DNA extension. This is approach first used by Shank et al., *Nature* 2010. The reference below was added to Supporting information (section Analysis of Unfolding and Refolding Trajectories):

- Shank EA, Cecconi C, Dill JW, Marqusee S, Bustamante C. The folding cooperativity of a protein is controlled by its chain topology. *Nature.* 2010 Jun 3;465(7298):637-40. doi: 10.1038/nature09021. Epub 2010 May 23. PMID: 20495548; PMCID: PMC2911970.

2. Line 154-155: may add a reference to the SI at the end of the sentence for the fitting equation.

We thank the reviewer for the comment and we have added at the end of the sentence 'Supplementary information'.

3. The x-axis label in Figure 3d-f denotes the waiting time for the refolding of the protein (i.e., refolding time). I suggest explicitly mentioning “refolding” in the axis label here (up to the authors).

We made the change in Figure 3d-f that the reviewer suggested.

4. Can authors define quantities such as density and fraction of states used in Figure 3?

We thank the reviewer for this comment as it provides additional clarification of these terms. In the legend of Figure 3 we added the definitions of density and fractions of states.

5. Line 180: “The analysis does not indicate which cluster incorporates FAD-bound and unbound dCRY states.” However, the section starting at Line 315 and Figure 5 do indicate that these intermediates can be mapped based on the cluster analysis. The authors may want to clarify this in line 180 or add a comment referring to the later section.

The reviewer is correct in that the clustering alone cannot determine which states incorporate FAD bound and unbound states, but later we provide a model based on the kinetic analysis that enabled us to eliminate potential states. We hope to have clarified the point the reviewer makes by editing and adding two sentences in that paragraph: “The cluster analysis does not identify whether FAD-bound and unbound dCRY states are incorporated in any of the five clusters. It is possible that all clusters include FAD-bound and unbound states, or just a few states bind the cofactor. However, the experimental data in Figure 3a provides evidence that eliminates possible states along the dCRY folding pathway.”

6. Line 190 in SI: in the second term inside the parenthesis, $k_{-2} \diamond k_{2}$

Thanks to the reviewer for identifying this mistake.

7. Lines 194 and 195 in SI: $[I_{\{F.FAD\}}]$ should be $[FAD]$ following scheme 1.

Thanks again to the reviewer for identifying these mistakes.

8. To analyze the extension measurements from optical tweezers, the authors used crystal structural information of dCRY. This methodology to use protein atomic structures to interpret the extension data from optical tweezers is previously published (Reban, AA et al, BJ, 110: 441, 2016), which may be cited.

We agree with the reviewer that this reference should be included. We have done so in the main text and supporting information.

Reviewer #3 (Remarks to the Author):

Through the help of optical tweezers and in-depth kinetic analysis, the present single-molecule study captured the unfolding-folding of the *Drosophila* cryptochrome (dCRY) with the help of the co-factor flavin adenosine diphosphate (FAD) and attempted to identify the mechanism for the same. Five distinct states were identified through which a kinetic scheme was developed and using statistical techniques, the kinetic rate constants were determined. The results and explanations are well-presented. Overall, the study provides important insights about the absolute need of FAD in allowing dCRY to attain a stable native state via experimentally observed distinct intermediate states supported by quantitative estimation of kinetic rate constants. The work maybe further refined after addressing some points.

1) Line 126: “Figure 1e” is mistyped as “Figure 1c”.

We thank the reviewer for identifying this mistake, it has been corrected.

2) Line 154: The details for the equation used to determine the apparent FAD dissociation constant should be provided.

We thank the reviewer for this comment, and we have added to the section Analysis of Unfolding and Refolding Trajectories in Supplementary Information the equation used to obtain the apparent FAD dissociation constant. We also slightly modified the main text to “Assuming that the fully folded protein is bound to FAD, we plotted the fraction of fully folded states as a function of the $\log[\text{FAD}]$ to obtain an apparent FAD dissociation constant: $K_{\text{dapp}} = 0.11 \pm 0.01 \text{ nM}$ (mean \pm standard error) (Figure 2b).”

3) From the discussion in the section “Contribution of FAD Moieties to dCRY Folding”, it is evident that isoalloxazine ring as well as the adenosine moiety are individually able to aid in complete folding of dCRY. What is the relative efficiency of each of them in promoting dCRY folding, especially with respect to their interactions with the protein? Do both these moieties follow the same pathway?

The point brought up by the reviewer is important, namely, what the efficiencies of the respective cofactors mean with respect to their interactions with the protein, and whether these moieties follow the same folding pathway.

We find that the isoalloxazine ring is the major contributor to complete folding of dCRY. Riboflavin (isoalloxazine + ribitol) or FMN (isoalloxazine + ribitol + 1 phosphate) promote complete folding with efficiencies of 39% and 85% of events, respectively. The latter value is similar to FAD. This is summarized in Figure 4b. Adenosine alone or with 1 or 2 phosphates is able to promote folding in 20% of events or less (Figure 4b) which are values significantly lower than those observed for riboflavin, FMN or FAD. Thus, the data suggests that each ring may follow either a different folding mechanism or that the kinetic rate constants are going to be slower for adenosine-containing rings given their lower folding efficiency.

Analysis of the structure of dCRY indicates that the isoalloxazine and adenosine rings establish interactions with different residues in dCRY (Figure 4a). These differing interactions suggest that the two moieties may produce different intermediates along the folding pathway, namely, it may be that the adenosine ring makes interactions with residues that are different from those that the isoalloxazine ring establish as the protein starts to fold. This could lead to new intermediates or to different rate constants.

However, with the single molecule data at hand, it is not possible to determine with high certainty whether the moieties containing isoalloxazine would follow the same or different intermediates as the moieties containing adenosine. This would require an entirely new study in which we perform kinetic refolding and binding experiments (as shown for FAD in Figure 3) for cofactors containing adenosine. This would allow a full characterization of intermediates (via clustering) and their kinetic rate constants (via global fitting).

Motivated by the reviewer's questions, we added a small discussion on this point at the end of the section **Contribution of FAD Moieties to dCRY Folding**: "Given that the adenosine ring establishes contacts with residues in dCRY that are different from the isoalloxazine ring (Figure 4a), it is possible that the folding pathway is going to be different, or the kinetic rates are going to be slower for adenosine-containing cofactors due to their lower folding efficiency."

4) Do different FAD moieties lead to separate folding mechanisms or is Scheme 1 universally followed?

With the data at hand, we cannot establish whether the different FAD moieties lead to different folding mechanisms. We discuss this in the previous question and added a short paragraph to the section **Contribution of FAD Moieties to dCRY Folding**. We believe that this will be a separate study where we perform similar experiments as in Figure 3 but with the different cofactors used in this study.

5) A comparative study between FAD binding to clusters I 1 and I 2 should provide significant insights about the mechanism of the folding process and could be explored.

We agree with the reviewer in that binding of FAD to clusters I1 and I2 provide insights into the mechanism of the folding process. In fact, given the comment made by the reviewer we have decided to divide the section "Mapping Folding and FAD-Bound Intermediates Along the dCRY Folding Pathway" in two parts. The first part will retain the same subtitle and includes the results from the worm-like chain analysis to infer the secondary structures involved in each section. The second part has a new title "FAD binds to dCRY Intermediates Following Two Different Mechanisms," where we discuss the possible two mechanisms by which I1 and I2 bind FAD. We believe that this new section enhances the idea that a complex protein like dCRY may include more than one mechanism to fold around the cofactor.

6) The color scheme for figure 3, the colors used for C3 and C4 are almost indistinguishable.

Thanks to the reviewer for the suggestion. We have changed the colors in Figure 3 to amplify their differences.

7) Did the authors determine the time for the protein to attain the native structure from the unfolded state? A statistical analysis of the folding time could provide insights to the stochasticity of the unfolding-folding process.

As the reviewer mentions, there is an underlying stochastic behavior of the refolding process. In our experiments, we were able to obtain data using a refolding time of up to 40 s, where we obtained a probability of 0.66 +/- 0.04 for dCRY to natively fold starting from the unfolded state and in the presence of saturating amounts of FAD ([FAD] ~ 100 Kd_{app}) (Figure 3f). Motivated by the reviewer's comment, we estimated the average first-passage time from the unfolded state to the natively folded structure bound to FAD. We followed the approach by Cleland in 1975 (Reference at the end of this answer). Briefly, we rewrote the net rate constants from the unfolded to the folded state as the original rate constants in Scheme 1 multiplied by the partition between forward and backward flux:

Where:

$$k'_7 = k_7; k'_6 = k_6 \cdot \frac{k'_7}{k_{-6} + k'_7}; k'_4 = (k_4 \cdot [FAD]) \cdot \frac{k'_6}{k_{-4} + k'_6}; k'_2 = k_2 \cdot \frac{k'_4}{k_{-2} + k'_4}; k'_1 = k_1 \cdot \frac{k'_2}{k_{-1} + k'_2}$$

The values of the rate constants were obtained from Supporting Table S2 (Global Analysis of Kinetic Refolding and FAD Titration Data). Using the net rates, the average first-passage time from U to F_{FAD} as a function of [FAD] is:

$$\tau_{U \rightarrow F_{FAD}} = \frac{1}{k'_1} + \frac{1}{k'_2} + \frac{1}{k'_4} + \frac{1}{k'_6} + \frac{1}{k'_7}$$

The plot of $\tau_{U \rightarrow F_{FAD}}$ vs. [FAD] shows that dCRY has an average first-passage time that asymptotically reaches 30 s at saturating concentrations of FAD. This value is consistent with our experimental observations showing 0.66 +/- 0.4 probability of folding to the native state when we wait 40 s and we use 100-fold excess of FAD relative to the Kd. In fact, the cumulative probability distribution of an exponential function using a time constant of 30 s indicates that the

probability of folding at 40 s is 0.73, a value that is in agreement to the observed probability of 0.66 +/- 0.4.

We have added to the main text this result and added to supplementary information the analysis of first-time passage based on the obtained kinetic rate constants from the global fitting. We think this strengthens the analytical framework used in this study.

- Cleland WW. Partition analysis and the concept of net rate constants as tools in enzyme kinetics. *Biochemistry*. 1975 Jul 15;14(14):3220-4. doi: 10.1021/bi00685a029. PMID: 1148201.

8) In Supporting Fig. S4 (a-b), please define “E” and “V” mentioned in the respective legends.

We thank the reviewer for this point, and we have added the definitions of “E” and “V” in the figure legend of Fig. S4 and in Supporting Information. Please note that we have removed the original Fig S4b and only kept S4a. This is because in this revised version we only use the BIC criterion in the clustering analysis.

Reviewer #4 (Remarks to the Author):

Foroutannejad et al have made an important experimental contribution towards understanding protein folding for large proteins with co-factors. It was very interesting to read.

The strength of the study is, in my opinion, the description of the folding process of dCRY and how it involves several steps with co-factor involvement. I found the data in figure 1c and 1d very impressive. And I like the authors attempt of describing these steps in the unfolding process, albeit I have some concerns (below). Another strength of the paper is the systematic investigation of different moieties of the cofactor, which I think was a great and thorough piece of work, and aid in a better understanding of the folding process of dCRY, as well as the role of the components of the co-factor.

I have some concerns regarding the clustering method:

- I am in doubt how much to trust the clustering algorithm. It would be great to see it tested against a known system (with real experimental or synthetic data), to benchmark if the method can reproduce the correct (known) number of clusters. This is probably a lot of work, but if such test/benchmarking cannot be presented, then I think the authors should be slightly more humble about their conclusions, number of clusters etc, and discuss potential limitation or pitfalls of the method.

The reviewer's point is important, namely, that the number of clusters identified may not account for weakly occupied states, and therefore our models and interpretations may provide an incomplete picture of the molecular process under investigation. A recommendation was to benchmark the approach with real or synthetic data. This is possible, but like the reviewer indicates, it is a significant effort. We therefore believe that indicating the potential limitations of the method is more appropriate for the scope of this manuscript. We therefore have added the following new text (3rd paragraph) in conclusions:

“Given the high degree of heterogeneity of the single molecule data, it is possible that the analysis used in this study missed transient states that have very low probabilities and therefore the number of clusters may be underestimated. Thus, the folding mechanism of dCRY may be more complex than what we have outlined. Nonetheless, the methods described herein allow for unprecedented quantification of key states in single molecule folding data that display such a high degree of complexity and heterogeneity.”

- The authors claim to use BIC and ICL. But these are to different model assessment equations (see e.g. <https://arxiv.org/pdf/1411.4257.pdf>), which do not necessarily result in the same number of clusters. Maybe the methods gave the same result for this dCRY (although, from figure SI-3 it is not clear that $n=4$ is a minimum or a maximum of ICL). The authors definitely have to describe this part more thoroughly and clearly. Why use both BIC and ICL? what if they give different results in the clustering - which one to choose then, and why?

The reviewer is correct that different model selection criteria may not lead to the identification of the same model. We were trying to show that the model selected using BIC and ICL were more or less concordant. We have removed the results of the ICL in the revised version and have only kept those based on BIC. Supporting Figure 4 now shows only the results based on BIC.

- A small, but general comment, is that bootstrapping is highlighted on several places in the manuscript, e.g., in abstract: "that integrates clustering, bootstrapping, and global fitting...". Bootstrapping is "just" a way to determine the errors, if I understand the presented method correctly, and should therefore, in my opinion, not be highlighted so much. It could just be mentioned in the method section.

The reviewer is correct that bootstrapping is a method for determining the uncertainty/error in estimated parameters. As suggested by the reviewer we have deleted bootstrapping from the Abstract, the Introduction and once in the Conclusions section. We now describe the bootstrap procedure in the Supplementary Information and refer to it sparingly in the main paper.

Following is a list of suggestions for improvements. Major points must be addressed before publication. These are marked with an asterisk "*"

title: "a light-sensing protein" is a bit unspecific

We have changed the title to "The cofactor-dependent folding mechanism of *Drosophila* cryptochrome revealed by single-molecule pulling experiments" to be more specific to the protein type.

line 15-17: it is an unpleasantly long starting sentence - any book on scientific writing will tell you that is a bad idea.

As suggested by the reviewer, we have divided this sentence in two.

line 16: I suggest you remove "the" in "the polypeptide", unless you refer to a specific polypeptide

We have removed "in" and replace it with "a" to make it more general.

line 16: why do you write "polypeptide" here, when you use the term "protein" or "multidomain protein" in the rest of the paper? Personally, I think "protein" is a better term.

The reason we use the word "polypeptide" in the (now) second sentence of the Abstract is that we want to maintain a broad expectation on how the cofactor interacts with dCRY, without implying that the protein is folded or unfolded. By writing "protein" at the start of the Abstract, we believe we imply that the protein is folded and then it binds to the cofactor.

line 20: "that binds FAD, one of the most common, complex cofactors" could be changed to: "that binds the cofactor FAD." Alternatively, please specify what you mean by "common" and "complex"

We thank the reviewer for making this point. When we wrote common and complex, we referred to commonly found cofactors despite being structurally and chemically complex. We have modified the sentence to: "...FAD, one of the most common, yet chemically and structurally complex cofactors found in nature."

line 23: "diffusion-limit" in two words: "diffusion limit"

We thank the reviewer and made the change.

line 26: "chemical" - isn't that word redundant?

Indeed, it is redundant, and we have made the deletion.

* line 30: "with structural data" - be more specific. what structural data?

We referred to the high-resolution structure of dCRY. We have added the words "high-resolution."

* line 32-34: generally useable - examples of other systems?

We understand that the reviewer refers to examples of other systems, and those would make a long list. We prefer to leave the sentence as it is, given that this study should be applicable to the folding of any protein system that is large, multidomain, or otherwise topologically complex and that binds cofactors. There are so many examples, it seems misplaced to us to suggest that a specific few are particularly suited to the method. Studying these systems has been a limitation in the field.

line 37-43: are refs 1-7 necessary?

We believe that references 1-7 are important. Since the solution of the first protein structures, the presence of cofactors (i.e., hemoglobin) has been acknowledged but its fundamental role in how to achieve its native state has been less explored. Moreover, there are several advancements in protein structure prediction or determination (experimentally), but how a protein achieves a final native state remains a major challenge.

* line 44: what is meant by "bulk studies"?

As opposed to experiments done at the single molecule level, like the one we use in this study with optical tweezers, "bulk studies" means studies in solution with relatively high concentrations of proteins. We believe this term is necessary in the text to highlight the advantage of using single molecule techniques as opposed to bulk solution methods. We have changed the phrase "bulk studies" to "studies in bulk," which have been used more commonly in biochemical and biophysical publications.

line 51-52: it would be interesting with a reference or argument for why FAD is "common" as well as a specification of what is meant by "large" in this context

We have included reference 9 in the section the reviewer mentions, as this review highlights different types of cofactors, from simple and small ones like a metal ion, to larger and more chemically and structurally diverse like flavins, hemes or FAD. FAD is a protein cofactor found in all kingdoms of life, which is why it is one of the most used in nature. In the Abstract we made changes to represent this point explicitly.

* line 53-54: what is meant by "in bulk"? do you mean in solution?

Yes, we thank the reviewer and we have replaced the word "in bulk" with "with bulk solution methods." We kept the word "bulk" in place to indicate that previous studies on folding of large proteins are lacking because of the high concentrations that most methods in bulk solution need.

line 54: does "it" here refer to single molecule techniques? then it should be plural

Thank you to the reviewer for pointing out this grammatical error. It has been corrected.

line 56: "importantly". In my opinion, the other points are equally important.

We have deleted the word 'importantly'.

line 52-57: this is a very long sentence, which makes it hard to read. Could you rephrase, or split in two?

As suggested by the reviewer, we have divided the original sentence in two.

line 58-72: some abbreviations could be avoided: CTT, CCM - are they necessary?

We prefer to keep the abbreviations like CTT and CCM. They are commonly referred in the field of light sensing proteins and recognized as functionally relevant structural motifs.

*Figure 1a: the structure in figure 1a (left) is too small, and it is difficult to link what is described in the text with the figure. E.g., it would be easier to follow, if the three loops were annotated on the structure. And it is hard to see the 4-helix bundle. This is a pity, as it is an important figure panel in order to understand the paper.

We thank the reviewer and have made the figure panel larger and identified the three loops and other elements in the figure itself.

*line 82: "structural data". Please, be more specific

We have added the words "high-resolution" and added the reference corresponding to the manuscript, reference 30 in the original list: Zoltowski, B. D. et al. Structure of full-length *Drosophila* cryptochrome. *Nature* 480, 396–9 (2011).

Figure 1b: no hyphen in Avi-tag and ybbR-tag. Annotations "CoA" and "B" are written with too small font. Why not write out "Biotin"? abbreviation SA and AD not explained.

We thank the reviewer for pointing out the missing information. We have deleted the hyphens in Figure 1b, added the word biotin and digoxigenin and explained in the same figure legend and main text what SA and AD mean.

line 96: please, write out "base pair" instead of abbreviating

We spelled out base pair as suggested by the reviewer.

line 97: a bit unclear what the 5' end of a double helix is... there is a 5' and a 3' in both ends.

Here it refers to the fact that the digoxigenin molecule is covalently linked only to the 5' end of the double stranded DNA. The 3' end is not modified.

line 100: what specifically do the authors refer to in SI here?

The reviewer is correct in that the reference to Supplementary Information was not necessary and therefore it was deleted.

*line 100-102: the spectroscopic data show that dCRY is functional, but not necessarily that the fold is unchanged.

The reviewer makes an important point. There are several studies that have used the tags ybbr and AVI without perturbing the fold of proteins (Reviewed here: Bustamante C, Alexander L, Maciuba K, Kaiser CM. Single-Molecule Studies of Protein Folding with Optical Tweezers. *Annu Rev Biochem.* 2020 Jun 20;89:443-470. doi: 10.1146/annurev-biochem-013118-111442. PMID: 32569525; PMCID: PMC7487275.)

Moreover, the fact that the modified dCRY functionally behaves like the wildtype protein (Supporting Figure 2a, b and c) lends confidence that the fold is not significantly changed. In fact, not only the steady state measurements show similar behavior (Supporting Figure 2a) but the kinetics of photoreduction and recovery are very similar. Lastly, based on the WLC model we find that the protein can fold entirely when it is bound to FAD. We think that these results indicate that the fold of the modified protein is very similar to that of the wildtype protein.

*Corrections to SI fig S2:

SI, fig S2a: axis labels needed.

SI, fig S2bc: what does A403 and A365 refer to?

why negative labels on a y-axis showing percentage? should it be 100% instead of 1% as max? in that case, the max y-value should be 100, not 105.

(and plotting in Excel - hmm... If everything was notated correctly etc, I wouldn't complain. But I can't help feeling the figure is a bit unfinished - not ready for a thesis, and definitely not for a peer-reviewed journal publication).

We thank the reviewer for making these annotations. We have corrected all the panels and made them in the same software. We have added panel 'd' in which we show that after extensive dialysis the cofactor FAD dissociates.

line 120: when uncertainty is 15, you should not give the mean with 4 significant figures. Just write 206 +/- 15. https://en.wikipedia.org/wiki/Significant_figures

As suggested by the reviewer, we have written '206.'

*line 120: just to be sure. Do you mean standard error = standard deviation of the mean. Or, the standard deviation for the N experiments? You should give the standard error as uncertainty on the mean.

We thank the reviewer for the clarification. It is the standard error of the mean and we have made the change in the paper.

line 122-123: please write the equation on its own line, for clarity

We have modified and simplified the text that the reviewer mentions and refer to the section Analysis of Unfolding and Refolding Trajectories in Supplementary Information for the details on the calculation. We believe that having this calculation as a stand-alone equation in the main text is not critical. Therefore, we report the expected value from the dCRY structure and refer to Supplementary Information.

line 123: I think it is optimistic to estimate this number with an accuracy that can justifying 4 significant figures. Probably writing 198 nm is fine.

We made the change in the main text as suggested by the reviewer. This change was made throughout the manuscript.

* line 126: figure 1c is not "in absence of the cofactor", but has a 25 μ M conc of FAD. Do you mean fig 1e? please correct or clarify.

We thank the reviewer for identifying the error. Indeed, we were referring to Figure 1e.

figure 1c-f: colors are inconsistent, if I read the figure correctly. left side: blue is unfolding, red is refolding. But right side: blue is with FAD, red is without FAD.

We made changes in Figure 1. We retained blue and red for unfolding and refolding in the molecular trajectory, which was the original color code. The points of ΔX_T corresponding to the WLC analysis in 1d and 1f are in black and grey, corresponding to conditions with and without FAD.

line 143: what is referred to in SI here?

We realized that the reference to Supporting Information was not needed in line 143.

* line 140: you write that the degree of folding is between 0 and 1, but clearly (from the figure), it can exceed 1. I do see the logic after reading a couple of times, but it should be more clearly written.

The reviewer is correct in that some experimental values exceed 1, and this is due to experimental uncertainty. We have modified this text to better explain how we initially interpret degree of folding. The new text reads: "The results were plotted as degree of folding, which was calculated by dividing the observed ΔLcT at each FAD concentration by 198 nm, the theoretical ΔLcT of full-length dCRY (Figure 2a, Supplementary Information on Analysis of Unfolding and Refolding Trajectories). Degree of folding of 0 reflect the unfolded state, whereas values ~ 1 reflect the fully folded state. Degree of folding between 0.1 and 0.9 were considered intermediate states with partially folded structures."

line 150: In my opinion, there is no need to refer to the figure again, you are in the middle of a discussion of it.

We have removed the reference to Figure 2a, as suggested by the reviewer.

*line 154: it is not a dissociation constant. calling it so is confusing - indicating that you have made a binding experiment.

The obtained value is an apparent dissociation constant and corresponds to the inverse of the apparent association constant between FAD and dCRY. We use the nomenclature of dissociation constant because it has the units of Molar (in this case nanomolar). This designation has the advantage that it provides the same units as the x-axis, and it is a common form to report affinities in biochemistry and biochemistry. To avoid any confusion, the equation has been added in Supplementary Information in Analysis of Unfolding and Refolding Trajectories.

line 150-152: that whole discussion of intermediates seems a bit redundant. The message is clear-cut without.

After revising the text, we agreed with the reviewer in that the description of intermediates in that section is redundant. Therefore, we deleted those sentences.

* figure 2: very nice data. But why do you want to do that division into (arbitrarily chosen) subdivisions? A plot of average degree of fold as function of concentration would summarise the findings without subdivisions, and in a more simple and reproducible way, instead of Fig 2b. Please change this, or justify your choices. Also, discuss the subdivisions. Why these? Would the results change if you chose other subdivisions? Would this subdivision work for other systems?

Thanks to the reviewer for this comment. The initial idea to separate the data in Figure 2 was based on the finding that the change in contour length for the folded protein was 206 ± 15 nm (mean \pm standard deviation). We made a first-order approach using the standard deviation to represent the error in the degree of folding analysis for the folded state. Because the standard deviation is close to 10% of the total change in contour, we conservatively use values of degree of folding > 0.9 as representative for the folded state. This subdivision is likely protein specific and instrument specific. We would not suggest that this subdivision can be used directly in other studies. Motivated by the reviewer's comment we have made three changes. We have removed the colored divisions in Figure 2, we have used the y-axis limits, and we have included the following statement in Supplementary Information (section Analysis of Unfolding and Refolding Trajectories) where we discussed this division: This division is based on the observed standard deviation of ΔLcT of the fully folded protein, which is 15 nm or 7 %. We rounded up the value to 10% as a first-order approximation to estimate the values of degree of folding for the folded state (> 0.9) and unfolded states (< 0.1). Anything in between are considered intermediate states.

fig 2a: please use the same ylimits for all conc - that would make it much easier to compare.

We have addressed this in the current manuscript version.

*fig 2b: something is wrong with the x-axis. $\log(0)$ is mathematically undefined, but you have a 0 M concentration. Where is that plotted? when you take the log of a conc, the units are not longer M, but $\log(M)$

We have addressed the comment made by the reviewer.

line 154-155: why give uncertainty as 95% CI (2 std dev, assuming normal distr), when you gave results with 1 sd uncertainty previously (and later) in the manuscript. It is not wrong, but a bit inconsistent.

Thanks to the reviewer for point this inconsistency. We have changed the value to standard error.

line 158: the multiple steps in the unfolding are, in my opinion, more clear from fig 1c than from fig 2a.

The reviewer is correct in that Figure 1c shows the unfolding intermediates more clearly. However, the degree of folding in Figure 2a reflect the total change in extension upon unfolding, and therefore the amount of protein that was folded before any unfolding step occurred. If the protein folded completely, the degree of folding will be ~ 1 but if only the protein folded up to 50%, then the degree of folding is ~ 0.5 . Therefore, the variability of the plot in Figure 2 indeed reflects various intermediates along the folding pathway.

figure SI-3: please use the same ylimits so histograms can be compared visually. How come you see any folding in 0 s? this should be explained more thoroughly

We have changed the plots to 1 s. This was a mistake on our side and did not reflect the plots in Figure 3d-f. We have changed the ylimits from 0 to 50 to all plots except time 1s at FAD = 0, which we left at 100 given that almost all values had zero degree of folding. Changing all other plots to 100 would make a visual inspection of all other data very difficult.

*line 181-182: "direct" is a bit of an overstatement. It is a rather indirect method. The data in figure 1c are, on the other hand, direct evidence of unfolding steps.

We have deleted the word "direct" from the sentence indicated by the reviewer.

*line 198-201: please provide p-values from the F-test in the main text. Please provide N, and describe what you have used as number of degrees of freedom.

We have created a new Supporting Table S3 where we provide a thorough statistical comparison from the fits of models 1, 2 and 3. We have added the p-values to the main text. As a reminder, model 1 follows Scheme 1 in which FAD binds to I1 and I2. In model 2, FAD binds to I1 and in model 3, FAD binds to I2. The table reports the estimated parameters, the degrees of freedom, as well as the F-test statistics and corresponding p-values for comparing the fit of the smaller models to the larger model. Altogether, model 1 provides the best fit to the combined FAD titration and kinetic refolding data.

figure 4: please denote riboflavin, ribitol and FMN on figure 4a.

We have done this as requested by the reviewer.

SI-Fig 4: unfolded stated -> unfolded states

this sentence is difficult to understand: "c- the standard error on the fractional populations was determined by the number of bootstrap iterations performed was increased"

We have changed the legend accordingly. We thank the reviewer for pointing this out.

SI-Fig 4c: standard error has units – please provide these

The plot in SI-Fig4c (now SI-Fig 4b) shows the standard error for one of the cluster means. It has no units because the data has been standardized (observed total change in contour divided by the value expected for the natively folded protein).

*SI-Fig 4: please write what “E” and “V” means

We included in the section of clustering in Supplementary Information the definition of E and V. We also included those definitions in the figure legend.

REVIEWERS' COMMENTS

Reviewer #1 (Remarks to the Author):

Foroutannejad et al. have satisfactorily addressed my questions and comments. Together with the modifications suggested by the other referees, the manuscript is substantially improved.

Hence, I recommend publication of their manuscript after the following minor changes have been made.

* p. 7, line 217: ' $k_3 = (2.8 \pm 0.3) \cdot 10^9$ ' should be altered to ' $k_3 = (2.8 \pm 0.4) \cdot 10^9$ ' (see Table S2: Global Analysis of Kinetic Refolding and FAD Titration Data).

* p. 7, line 221: ' $k_5 = (0.12 \pm 0.04 \text{ s}^{-1})$ ' should be altered to ' $k_5 = (0.20 \pm 0.04 \text{ s}^{-1})$ ' (see Table S2: Global Analysis of Kinetic Refolding and FAD Titration Data).

* p. 7 line 237: '... which range between 8-270 nM' should be altered to : '... which range between 8-240 nM' (see SI of ref 42).

Reviewer #2 (Remarks to the Author):

The authors have carefully revised the manuscript and addressed my comments. I support the publication of this revised version of manuscript.

Reviewer #4 (Remarks to the Author):

The authors have addressed all concerns (and there were quite a few!) satisfyingly.

Dear Reviewers:

We thank you for taking the time to read our revised manuscript. Below are three minor changes observed by Reviewer 1. We have made the changes to the final version of the manuscript.

* p. 7, line 217: ' $k_3 = (2.8 \pm 0.3) \cdot 10^9$ ' should be altered to ' $k_3 = (2.8 \pm 0.4) \cdot 10^9$ ' (see Table S2: Global Analysis of Kinetic Refolding and FAD Titration Data).

The change was made, and it is on page 6, line 205.

* p. 7, line 221: ' $k_5 = (0.12 \pm 0.04 \text{ s}^{-1})$ ' should be altered to ' $k_5 = (0.20 \pm 0.04 \text{ s}^{-1})$ ' (see Table S2: Global Analysis of Kinetic Refolding and FAD Titration Data).

The change was made, and it on page 6, lines 209 and 210.

* p. 7 line 237: '... which range between 8-270 nM' should be altered to : '... which range between 8-240 nM' (see SI of ref 42).

The change was made, and it on page 7, line 225.